# Semi-empirical forecast modelling of rip-current and shore-break wave hazards

Bruno Castelle[1], Jeoffrey Dehez[2], Jean-Philippe Savy[3], Sylvain Liquet[4], and David Carayon[2]

[1]Univ. Bordeaux, CNRS, Bordeaux INP, EPOC, UMR 5805, F-33600 Pessac, France
[2]INRAE Nouvelle Aquitaine, Cestas-Gazinet, France
[3]SMGBL, Messanges, France
[4]Météo-France, Toulouse, France

**Correspondence:** Bruno Castelle (bruno.castelle@u-bordeaux.fr)

**Abstract.** Sandy beaches are highly attractive but also potentially dangerous environments for those entering the water as they can be exposed to physical hazards in the surf zone. The most severe and widespread natural bathing hazards on beaches are rip currents and shore-break waves, which form under different wave, tide and morphological conditions. This paper introduces two new, simple, semi-empirical rip-current and shore-break wave hazard forecast models. These physics-informed models, which depend on a limited number of free parameters, can be used to compute the time evolution of the rip current flow speed $V$ and shore-break wave energy $E_{sb}$. These models are applied to a high-energy meso- macro-tidal beach, La Lette Blanche, in southwest France where intense rip currents and shore-break wave hazards co-exist. Hourly lifeguard-perceived hazards collected during the patrolling hours (from 11AM to 7PM) during July and August of 2022 are used to calibrate the two models. This data is also used to transform $V$ and $E_{sb}$ into a 5-level scale from 0 (no hazard) to 4 (hazard maximized). The model accurately predicts rip-current and shore-break wave hazard levels, including their modulation by tide elevation and incident wave conditions, opening new perspectives to forecast multiple surf-zone hazards on sandy beaches. In addition, daily-mean hazard forecasts demonstrate even greater predictive skill, which is important for conveying straightforward messages to the general public and lifeguard managers. The approach presented here only requires a limited number of beach morphology metrics, and allows the prediction of surf-zone hazards on beaches where wave and tide forecasts are available.

## 1 Introduction

Predicting natural hazards, such as flash floods, wildfires, and hurricanes, and disseminating warnings based on those predictions is crucial to protect property and natural resources, but also to protect people from injury and death (National Research Council, 1991; Merz et al., 2020; Bates et al., 2021). Over the last decades, prediction capabilities of atmospheric and hydrologic hazards, often referred to as weather-related natural hazards, have greatly increased (e.g. Brunner et al., 2021). While a lot of scientific effort and media coverage involve hurricanes (Gall et al., 2013), coastal flooding (Stockdon et al., 2023) or flash floods (Corral et al., 2019), in comparison less attention has been paid to the surf-zone hazards beachgoers expose themselves to. However, in the USA, rip currents on surf beaches were the third-leading cause of weather-related deaths from 2012 to 2021 according to the National Weather Service (US Department of Commerce), not far behind heat waves and flooding. Contrary to

most of these other weather-related natural hazards (e.g. Zscheischler et al., 2020), surf-zone hazards are not necessarily related to extreme events as fatal drowning and severe injuries at the beach predominantly occur during fair weather conditions, i.e. typically during warm, sunny and light-wind days (Dwight et al., 2007; Ibarra, 2011; Coombes et al., 2009; de Korte et al., 2021; Castelle et al., 2024). Therefore, improving our predicting capacity of surf-zone hazards on beaches is critical to reduce the burden of fatal drownings (Dusek and Seim, 2013) as well as that of other types of injuries.

Sandy beaches offer abundant recreational opportunities, tourism potential, and valuable ecosystem services (Ghermandi and Nunes, 2013; Hall and Page, 2014; Bujosa et al., 2015; West, 2019), including activities such as bathing and wading (Britton et al., 2018; Wood et al., 2022; Dehez and Lyser, 2024). However, beachgoers may face physical hazards within the surf zone. Among the most significant and widespread natural hazards leading to surf-zone injuries (SZIs), including drowning incidents, are rip currents (MacMahan et al., 2006; Dalrymple et al., 2011; Castelle et al., 2016b; Houser et al., 2020) and shore-break waves (Chang et al., 2006). Rip currents are narrow, seaward-flowing currents that originate in the surf zone, often near the waterline, extending through the breakers and sometimes beyond. These currents are a primary cause of unintentional drownings on many surf beaches worldwide (e.g. Brighton et al., 2013; Arozarena et al., 2015; Barlas and Beji, 2016; Li, 2016; Castelle et al., 2018; Brewster et al., 2019), as they can carry bathers offshore into deeper water, leading to drowning through exhaustion or panic (Brander and Short, 2001; Drozdzewski et al., 2012). Rip currents are driven by depth-induced breaking wave energy dissipation, although their formation mechanisms can vary (Castelle et al., 2016b). The most common rip type on intermediate beaches (Wright and Short, 1984; Castelle and Masselink, 2023) flows through channels carved into nearshore sandbars (e.g. Houser et al., 2013). These channel rips are caused by alongshore variations in breaking wave energy dissipation due to alongshore-variable sandbar depths (Bowen, 1969; Haller et al., 2002; Bruneau et al., 2011). Rip current activity typically increases with shore-normal wave incidence, higher wave height, longer wave period (e.g. Austin et al., 2010; Drønen et al., 2002; Bruneau et al., 2011; Winter et al., 2014; MacMahan et al., 2006), and lower tide level (Aagaard et al., 1997; MacMahan et al., 2005; Brander and Short, 2001; Houser et al., 2013).

Shore-break waves, in contrast, are plunging or dumping waves that break close to the shore on steep beach faces, causing a wide range of injuries, including severe spinal injuries (Chang et al., 2006; Robbles, 2006; Puleo et al., 2016; Castelle et al., 2018; Griepp et al., 2022). Most injuries associated with shore-break waves result from wave-induced impacts, followed by shallow water diving incidents, the latter often involving surfers (Thom et al., 2022). Unlike rip currents, research on shore-break waves is limited, mostly due to the challenges of quantifying their energy and impact forces on the human body. Castelle et al. (2024) observed that lifeguards in southwest France perceive shore-break wave hazards to be greater during long-period, near shore-normal waves and higher tides. Furthermore, prior studies indicate that the occurrence of spinal injuries from shore-break waves increases with long-period waves, and higher water level as waves typically break on the steepest sections of the beach (Castelle et al., 2019).

Despite our increased understanding of rip current dynamics, a limited number of rip current hazard forecast systems have been developed over the last decade. The approaches include for instance process-based modelling (Austin et al., 2013; Stokes et al., 2024, which requires detailed information of the beach morphology); statistical modelling of the likelihood of hazardous rip current using either lifeguard estimation of rip flow speed (Dusek and Seim, 2012, 2013) or measured rip-flow speed

(Moulton et al., 2017a); physics-based parametrisation of channel rip flow speed (Casper et al., 2024); hazard levels based on thresholds in tide elevation, wave height and period relying on lifeguard incident data (Scott et al., 2014, 2022). While some of these models skillfully predict rip current hazard levels, they have been validated on a limited number of beaches. Additional rip flow speed and/or lifeguard-perceived and/or topo-bathymetric datasets therefore need to be collected. In addition, these surf-zone hazard models only consider rip currents, while on some beaches the most threatening hazard is shore-break waves (e.g. Puleo et al., 2016). This calls for more generic surf-zone hazard models to be applied to a wide range of sandy beaches.

In this contribution, we present two simple semi-empirical rip-current and shore-break wave hazard forecast models which are validated at a high-energy sandy beach in southwest France, where strong channel rip currents and hazardous shore-break waves co-exist (Castelle et al., 2024) and are largely the most important cause of SZIs (Castelle et al., 2018). In Section 2 the field site and the 2-month dataset of environmental conditions and lifeguard-perceived hazard data used for model calibration are presented. Section 3 explains the development of the rip-current and shore-break wave hazard models. Results are given in Section 4, which are further discussed in Section 5. We show that the two models skillfully predict the lifeguard-perceived rip-current and shore-break wave hazards, including their complex modulation by tidal elevation, incident wave energy and neap-spring tide cycles. These simple semi-empirical models providing quantitative estimates of rip-flow speed and shore-break wave energy, and an associated 5-level scale hazard rating, only require a limited number of time-invariant free parameters related to beach morphology and wave breaking onset. These parameters can either be given based to some knowledge of the beach morphology, or through calibration using lifeguard-perceived hazard data. The proposed framework, here applied to a single beach in southwest France, offers new opportunities for forecasting rip-current and shore-break wave hazards at surf beaches with available wave and tide predictions.

## 2 Field site and data

### 2.1 La Lette Blanche Beach

La Lette Blanche beach (Figure 1b) is representative of the majority of open coast beaches in southwest France. Its typical beach state is intermediate and double-barred, with crescentic patterns on the inner intertidal bar and a transverse bar and rip morphology on the outer subtidal bar. The spacing between inner-bar rip channels is on average approximately 400 m. It is a meso-macrotidal environment, with an average tidal range of 2.6 m and a maximum of 4.4 m. It is exposed to high-energy ocean waves generated in the North Atlantic, with a summer-mean (July-August) significant wave height $H_s$ of about 1.1 m and a peak wave period $T_p$ of 9 s. Like other open beaches in the region, rip currents are ubiquitous (Figure 1c), with strong channel rips flowing through the inner-bar rip channels (Bruneau et al., 2009). Rip current activity peaks around mean low tide level under energetic, shore-normal wave conditions (e.g. Bruneau et al., 2011), which coincide with a higher occurrence of drowning incidents and rescues in southwest France (Castelle et al., 2019; de Korte et al., 2021; Castelle et al., 2024). Additionally, a significant number of mild to severe injuries in the surf zone are caused by shore-break waves (Figure 1c, Castelle et al., 2018). Research has shown that these injuries are more frequent during higher water levels and large tidal ranges when waves break over the steepest sections of the beach profile (Castelle et al., 2019, 2024).

La Lette Blanche beach is monitored by lifeguards during the summer months (July and August) between 11 AM and 7 PM. During these hours, a supervised bathing zone, typically less than 100 m wide, is established between two red and yellow flags (Figure 1b). This zone is strategically located away from potential rip currents. Due to the large tidal range, which causes rapid changes in the location, intensity, and nature of surf zone hazards, lifeguards may relocate the supervised bathing zone multiple times throughout the day. To communicate surf zone hazards, lifeguards use a color-coded flag system that reflects their assessment of conditions, including rip currents and shore-break waves: (1) a green flag indicates supervised bathing with no significant physical hazard; (2) a yellow-orange flag signifies dangerous but supervised bathing; and (3) a red flag means bathing is prohibited.

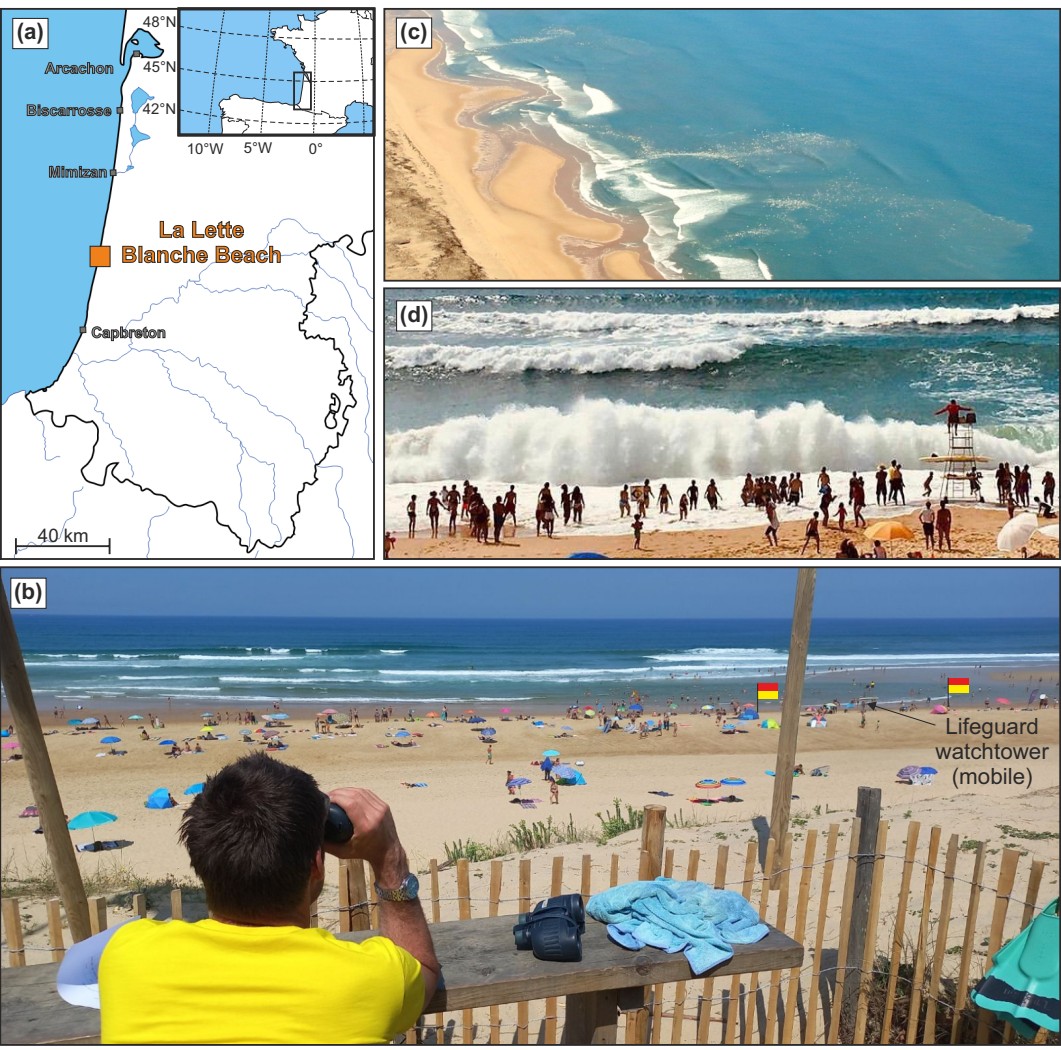

**Figure 1.** (a) Location map of La Lette Blanche beach, southwest France and (b) view from the lifeguard station on the top of the dune on July 14, 2022, 12PM (Ph. B. Castelle). Photographs in southwest France of the two major surf-zone hazards with (c) rip currents (Ph. Observatoire de la Côte de Nouvelle-Aquitaine, OCNA) and (d) shore-break waves (Ph. Syndicat Mixte de Gestion des Baignades Landaises, SMGBL).

## 2.2 Summer 2022 field experiment

During the boreal summer of 2022, from July 1 to August 31, a beach safety field experiment was carried out at La Lette Blanche beach. This study generated a unique multidisciplinary database encompassing various aspects such as beachgoer surveys, surf-zone drifter measurements, topographic surveys, lifeguard assessments of surf-zone hazards and beach crowds, as well as monitoring of environmental conditions. For further details on these datasets, please refer to Dehez et al. (2024) and Castelle et al. (2024).

In the present contribution, we use only lifeguard-perceived surf-zone hazards and wave and tide conditions. Given that the aim of the present contribution is to eventually operate surf-zone hazard forecasts, in contrast with Castelle et al. (2024) we used a numerical wave hindcast instead of in situ offshore wave measurements. The MFWAM (Météo-France Wave Model) based on the spectral wave model WAM (WamdiGroup, 1988) is the French version of the European Centre for Medium-Range Weather Forecasts (ECMWF) WAM model used by Météo-France for operational sea state forecasting, with a 0.1° grid resolution in the northeast Atlantic. It forces a high-resolution WaveWatch 3 wave model (Tolman et al., 2002), forced by winds from the ARPEGE model of Météo-France. The model uses an unstructured grid (Roland and Ardhuin, 2014), allowing the French Atlantic coast to be described with a resolution of approximately 200 m, with mesh size increasing to approximately 10 km at the boundary of the model a few hundred of kilometres offshore. Different coastal processes are represented in this model, such as unified parameterization of wave breaking from offshore to coast, wave reflection at the coast, refraction due to currents and bathymetry, and bottom friction. Modelled wave conditions were extracted in approximately 10-m depth in front of La Lette Blanche beach, i.e. to estimate the wave conditions outside of the surf zone. The data was further compared with the wave measurements at the directional wave buoy located approximately 80 km further north in 50-m depth, which in previous work was assumed to be representative of the wave conditions given the overall open and straight nature of the coast. Over the period from July 1 to August 31 of 2022, results show a root-mean-square error (RMSE), coefficient of determination ($r^2$) and bias of 0.17 m, 0.91 and -0.03 m. These metrics provide confidence into both model skill and the relevance of the wave buoy measurements used in previous work. In addition, tide conditions at the beach were estimated using the TPXO9 (version 5) 1/30°-resolution atlas (Egbert and Erofeeva, 2002) at the grid point the closest to La Lette Blanche beach. Figure 2a-d displays the wave and tide conditions during the 2022 experiment, showing significant wave height $H_{s0}$ (peak wave period $T_p$) ranging from 0.30 m to 2.19 m (3.85 s to 19.38 s) with a mean of 0.96 m (9.21 s). Waves were predominantly from the west-northwest, with the average angle of wave incidence with respect to shore-normal of 23.39°. Nearly 2.5 neap-spring tide cycles were covered (Figure 2d), with the daily tide range ($TR$) ranging 1.39–4.06 m with a mean of 2.73 m.

During each patrolled day of the summer 2022 beach safety experiment, the chief lifeguard (or the co-chief on the chief lifeguard's days off, two days a week) provided hourly estimates of rip current hazard ($RHl$) and shore-break wave hazard ($HSl$). These hazards were rated on a 5-level scale ranging from 0 (no hazard) to 4 (maximum hazard). Lifeguards were instructed to assess the environmental hazard level rather than the risk, meaning the focus was on the inherent hazard conditions rather than the likelihood of water users exposing themselves to rip currents or shore-break waves. Figure 2e,f illustrates the time series of daily-mean lifeguard-estimated rip current hazard ($\overline{RHl}$) and shore-break wave hazard ($\overline{SHl}$). The data indicate that the daily average rip current hazard generally increases with larger, longer-period, and near shore-normal waves. In contrast, the shore-break wave hazard is heightened under conditions of long-period, near shore-normal waves and large tidal ranges (Castelle et al., 2024).

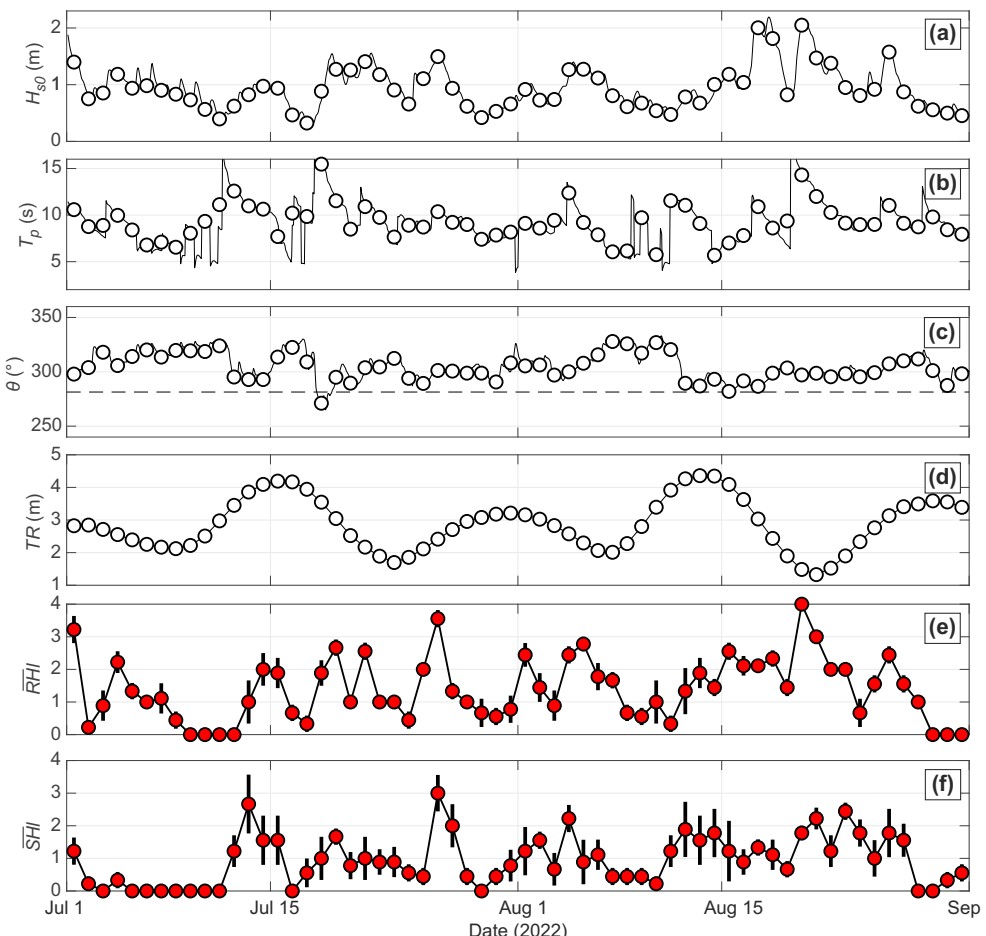

**Figure 2.** Time series of environmental conditions in nearly 10-m depth offshore of the study site and lifeguard-estimated surf-zone hazards during patrolling hours (11AM-7PM): (a) significant wave height $H_{s0}$ in 10-m depth; (b) peak wave period $T_p$; (c) angle of wave incidence $\theta$; (d) tide range $TR$; daily-mean (e) lifeguard-perceived rip-current hazard $\overline{RHl}$ and (f) lifeguard-perceived shore-break-wave hazard $\overline{SHl}$. In all panels the circles indicate the daily mean, and in (e,f) the vertical lines indicate the daily standard deviation.

## 3  Physic-based hazard models

### 3.1  Rip current

Rip current hazard can be estimated through the rip flow speed. Here we consider an idealised rip-channelled beach on which breaking waves drive a rip current through the deeper channels (Figure 3). Channel rips are essentially driven by the alongshore variation in breaking-wave-energy dissipation due to the alongshore variability in depth between the sandbars and intervening drainage channels. This can be simplified into the alongshore pressure gradients in the surf zone $d\bar{\eta}/dx$, with $x$ the longshore coordinate and $\bar{\eta}$ the wave set-up i.e. the increase in mean water level driven by wave breaking. These alongshore pressure

gradients drive feeder currents converging at the channels and turning offshore as rip currents (Haller et al., 2002). There is a
145 wealth of empirical formulas derived from field and laboratory measurements to estimate wave set-up (Gomes da Silva et al.,
2020). A popular, simple, formula gives the wave set-up at the shoreline as a function of the significant wave height upon
breaking $H_s$ only (Guza and Thornton, 1981; Raubenheimer et al., 2001; Atkinson et al., 2017):

$$\overline{\eta} \approx 0.16 H_s \tag{1}$$

The rip-current flow is controlled by the alongshore pressure gradient between the wave set-up immediately onshore of
150 the bar/rip system, in the alignment of the bar $\overline{\eta}_b$ and of the channel $\overline{\eta}_c$ (Figure 3c). Considering Equation (1), but looking
immediately onshore of the bar/rip system instead of the waterline, where the entire incident wave energy has been dissipated,
and by further ignoring set-down, wave refraction, wave-current interaction, we can make the first-pass assumption that wave-
set up immediately onshore of the bar/rip system is controlled by the change in wave height due to depth-induced breaking
across the bar and/or the channel. We can therefore assume $\overline{\eta}_b = 0.16\Delta H_{sb}$ and $\overline{\eta}_c = 0.16\Delta H_{sc}$, where $\Delta H_{sb}$ and $\Delta H_{sc}$ are
155 the decrease in wave height due to depth-induced breaking across the bar and the channel, respectively (Figure 3b). Note that in
the present work, the significant wave height $H_{s0}$ in 10-m depth was transformed into significant wave height at breaking $H_s$
using the direct formula of Larson et al. (2010). This formula allows to compute the incipient breaking wave properties based
on a simplified solution of the wave energy flux conservation equation combined with Snell's law, assuming shore-parallel
depth iso-contours.
Critical to both $\Delta H_{sb}$ and $\Delta H_{sc}$ is the depth-induced breaking wave height decay law. Unlike regular waves, there is no
simple method to estimate irregular wave heights in the surf zone, even on planar beaches. Previous studies (Dally, 1990)
have shown that the root mean square wave height distribution in the surf zone on planar beaches depends on various factors,
including beach slope and wave steepness. However, by neglecting wave shoaling effects and for the sake of simplicity, a
physics-informed (Dally, 1990) estimation of the depth-induced breaking significant wave height decay, $\Delta H_s$, for irregular
waves (Figure 3d), can be expressed as:

$$\Delta H_{si} = (H_s - \gamma h_i)^2 / H_s^2 \tag{2}$$

for $h_i > 0$ and $H_s > \gamma h_i$ (broken waves), where $h_i$ is the local water depth with subscript $i$ referring to the bar ($i = b$) or
the channel ($i = c$), $\gamma$ is the breaker parameter for random waves, and $H_s$ is the significant wave height at breaking (after
transformation through Larson et al., 2010). The depth-induced breaking significant wave height decay over the sandbar $\Delta H_{sb}$
(the channel $\Delta H_{sc}$) are given by:

$$\Delta H_{sb} = (H_s - \gamma(z_{bar} + \zeta))^2 / H_s^2 \tag{3}$$

$$\Delta H_{sc} = (H_s - \gamma(z_{bar} + \zeta + d))^2 / H_s^2 \tag{4}$$

with $\zeta$ the tide elevation, $z_{bar}$ the elevation of the sandbar and $d$ the channel depth (Figure 3b).

Following Moulton et al. (2017b), we assume that the ratio of bottom stress to the advection term is small, and that the balance of pressure gradients and advection along a streamline can be approximated using the Bernoulli equation. By further neglecting the effects of inertia in a longshore current driven by obliquely incident breaking waves, the rip flow velocity $V$ can be approximated as:

$$V \approx \sqrt{2g(\overline{\eta}_b - \overline{\eta}_c)} \qquad (5)$$

where $\overline{\eta}_b = 0.16\Delta H_{sb}$ and $\overline{\eta}_c = 0.16\Delta H_{sc}$ the wave set-up onshore of the bar and of the channel, respectively. Note that, because of the irregular wave height decay law (Equation (2)), the alongshore gradient in wave set-up, and thus rip-flow speed $V$, depend on $d$, $z_{bar}$ and $H_s$, whereas assuming regular waves, it would be independent of $H_s$ when depth-induced breaking occurs both over the channel and the sandbar.

This simple rip-flow model proceeds as follows : at each time step $t$, rip flow speed $V(t)$ is computed as a function of
$\overline{\eta}_b(t)$ and $\overline{\eta}_c(t)$, based on the significant wave height at breaking $H_s(t)$ and the local water depth across the bar (channel) $h_b(t) = z_{bar} + \zeta(t)$ ($h_c(t) = z_{bar} + d + \zeta(t)$), with $\zeta(t)$ the tide elevation, $z_{bar}$ the elevation of the sandbar and $d$ the channel depth (Figure 3b). The rip flow model $V$ includes only three time-invariant free parameters that need to be calibrated and/or inferred from field data : the breaker index for random waves $\gamma$, the sandbar elevation $z_{bar}$ and the channel depth $d$.

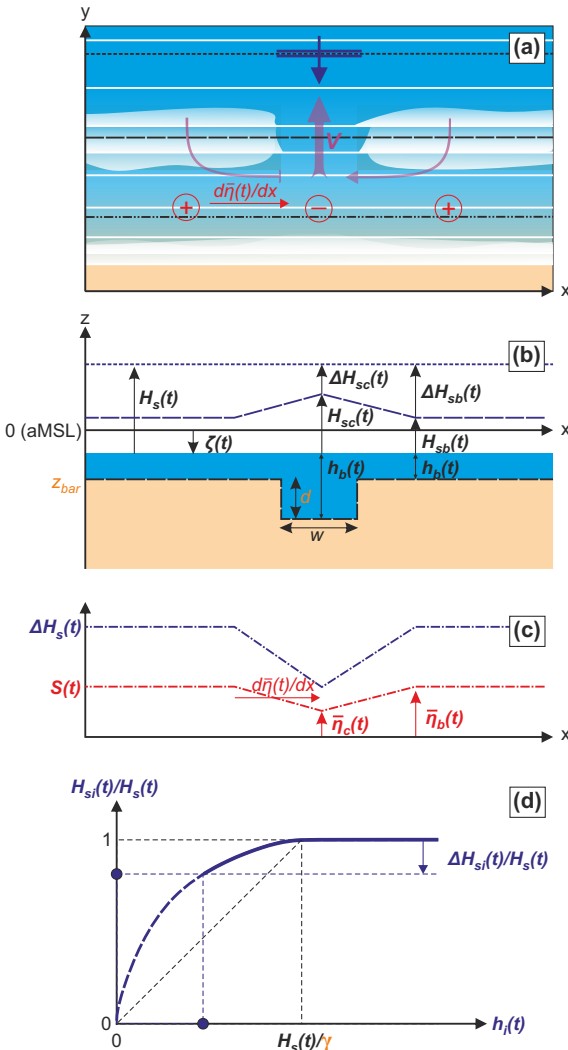

**Figure 3.** Schematics of the semi-empirical rip-current flow model: (a) top-view schematics of a rip-channelled beach with a rip current flowing through the deeper channel driven by the depth-induced wave breaking alongshore pressure gradients $d\bar{\eta}/dx$; (b) alongshore section with significant wave height at breaking $H_s$, across the bar $H_{sb}$ and across the channel $H_{sc}$, and their corresponding decrease with respect to breaking $\Delta H_{sb}$ and $\Delta H_{sc}$; (c) corresponding alongshore-variable wave set-up $\bar{\eta}$ across the bar $\bar{\eta}_b$ and across the channel $\bar{\eta}_c$, and resulting alongshore pressure gradient $d\bar{\eta}/dx$; (d) idealised, physics-informed, significant wave height decay model $\Delta H_{si}$, with subscript $i$ referring to the bar ($i = b$) or the channel ($i = c$), for a given significant wave height at breaking $H_s$, water depth $h_i$ with $\gamma$ the breaker parameter. In all panels, the time-invariant free model parameters are indicated in orange.

## 3.2 Shore-break

We used a similar, simple, semi-empirical approach to estimate shore-break wave hazard. Contrary to rip flow speed there is no theoretical framework to estimate a measure of the shore-break wave energy. The presence of shore-break waves can be estimated through the dimensionless Irribarren parameter $Ir$ which is a proxy for breaker type (Battjes, 1974):

$$Ir = \tan\beta / \sqrt{H_{sb}/L_0} \tag{6}$$

where $L_0 = gT_p^2/2\pi$ is the deep water wavelength, $\tan\beta$ is the local beach slope and $H_{sb}$ is the significant wave height at the 195   sandbar, i.e. upon shore breaking. While breaking goes from spilling to collapsing through plunging as $Ir$ increases, it does not provide information of the power of the breaking waves. Therefore, we introduce a shore-break wave energy parameter $E_{sb} = IrH_{sb}^2$, assuming no change in peak wave period as the waves pass over the sandbar(s) before reaching the shore, which therefore reads :

$$E_{sb} = H_{sb}^{3/2} T_p \tan\beta \sqrt{\frac{g}{2\pi}} \tag{7}$$

In order to compute $E_{sb}$, a beach profile and a wave height model is required. Here we consider an idealised Dean-like profile given by $z = 5 + ax^b$, which together with the tide elevation is used to compute the beach slope $\tan\beta(\zeta)$ (Figure 4a). Note that here we did not consider a Dean profile ($b = 2/3$) because we are interested in the intertidal, potentially bermed, part of the beach profile. Critical to $E_{sb}$ is the shore-break wave height $H_{sb}$. Similar to all intermediate beaches, the beaches in southwest France are barred, with depth-induced breaking wave energy dissipation across the offshore sandbar limiting the 205   breaking wave height at the shore, especially for lower tides. The sandbar was mimicked by assuming a terrace with a given elevation $z_{bar}$ (Figure 4a) where waves may dissipate before reaching the shore. Therefore, consistent with the rip current model, the wave height decay $\Delta H_s = H_s - H_{sb}$ was determined through the same simple wave height decay law (Figure 4b).

    This simple shore-break wave energy model proceeds as follows : at each time step, the beach slope $\tan\beta(\zeta(t))$ and the shore-break wave height $H_{sb}(t)$ are computed. If $\zeta(t) < z_{bar}$ during the lower tides, the sandbar is emerged and all the wave 210   energy is dissipated offshore, meaning $H_{sb}(t) = 0$. At the other end of the spectra, if $H_s(t) > \gamma_s(\zeta(t) - z_{bar})$ with $\gamma_s$ the breaker parameter for random waves of the shore-break model, there is no wave breaking across the terrace and $H_{sb}(t) = H_s(t)$. In between, offshore wave breaking occurs resulting in a decreased shore-break significant wave height by $\Delta H_s$ (Figure 4b), following the same depth-induced breaking irregular wave height decay law as for the rip current model, resulting in:

$$H_{sb}(t) = H_s(t) - \frac{[H_s(t) - \gamma_s(\zeta(t) - z_{bar})]^2}{H_s(t)} \tag{8}$$

The shore-break wave energy model includes four time-invariant free model parameters : $a$ and $b$ describing the beach profile shape, as well as $\gamma_s$ the breaker parameter for random waves and $z_{bar}$ the terrace/sandbar elevation.

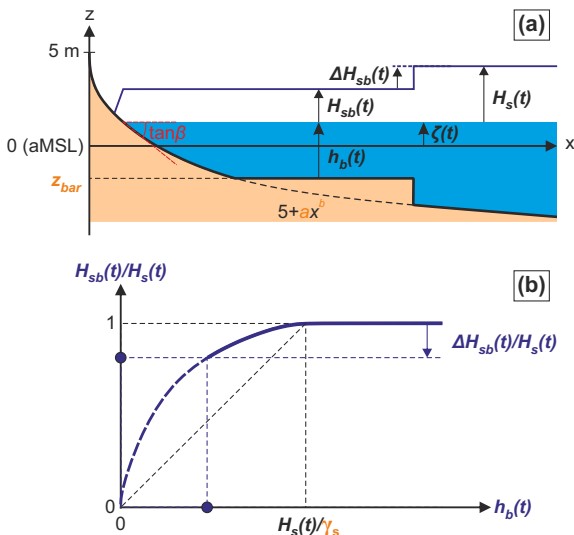

**Figure 4.** Schematics of the semi-empirical shore-break wave energy model: (a) Idealised beach profile based on a Dean-like profile $z = 5 + ax^b$ and a superimposed bar at $z = z_{bar}$, resulting in the shore-break significant wave height $H_{sb}$ that depends on (b) idealised, physics-informed, significant wave height decay model $\Delta H_{sb}$ for a given significant wave height at breaking $H_s$, water depth $h_b(t) = \zeta(t) - z_{bar}$ with $\gamma_s$ the breaker parameter for random waves of the shore-break energy model. In all panels, the time-invariant free model parameters are indicated in orange.

### 3.3 Model calibration and transformation into a 5-level scale hazard

A two-step approach was used, both steps using the lifeguard-perceived surf-zone hazard data $RHl$ (rip current) and $SHl$ (shore-break wave): (1) calibration of the free parameters of $V$ and $E_{sb}$ and (2) a quantile-quantile approach to transform
$V$ and $E_{sb}$ into a 5-level scale hazard. First, a large set of simulations were run for a wide range of free parameters. The optimal parameters were found by maximizing the coefficient of determination $r^2$ between $V$ ($E_{sb}$) and $RHl$ ($SHl$) during patrolling hours from 11AM to 7PM during the entire summer of 2022. Second, the values of $V$ and $E_{sb}$ concurrent to lifeguard observations were sorted and thresholds were computed in order to obtain the same number of modelled hazard levels (Table 1). Based on these thresholds in $V$ and $E_{sb}$, the complete time series of $V$ and $E_{sb}$ were transformed into modelled rip-current
($RHm$) and shore-break wave ($SHm$) hazard on the same 5-level scale as for lifeguard observations. The accuracy of these predictors was further addressed through confusion matrices. In addition, the modelled daily-mean rip-flow speed $V$ (shore-break wave energy $E_{sb}$) and the modelled daily-mean rip-current hazard $\overline{RHm}$ (shore-break wave hazard $\overline{SHm}$) were also compared with daily-mean lifeguard-perceived rip current hazard $\overline{HRl}$ (shore-break wave hazard $\overline{HRl}$) in order to address the ability of the two models to predict high-hazard days.

**Table 1.** Number of hourly lifeguard-perceived hazard observation $n$ (rip current : $RHl$, shore-break wave : $SHl$) discriminated by level (from 0 to 4) over a total of 558 hourly observations, together with the corresponding range of $V$ and $E_{sb}$.

| Hazard level | $n(RHl)$ | $V$ range (m/s) | $n(SHl)$ | $E_{sb}$ range (m$^2$) |
|:---:|:---:|:---:|:---:|:---:|
| 0 | 146 | $V < 0.31$ | 282 | $E_{sb} < 1.76$ |
| 1 | 167 | $0.31 \leqslant V < 0.91$ | 105 | $1.76 \leqslant E_{sb} < 2.93$ |
| 2 | 144 | $0.91 \leqslant V < 1.38$ | 102 | $2.93 \leqslant E_{sb} < 5.17$ |
| 3 | 83 | $1.38 \leqslant V < 1.89$ | 56 | $5.17 \leqslant E_{sb} < 8.67$ |
| 4 | 18 | $V \geqslant 1.89$ | 13 | $E_{sb} \geqslant 8.67$ |

## 4   Results

### 4.1   Rip current hazard modelling

The best pearson correlation ($R = 0.77$) between the modelled rip-flow speed $V$ and the hourly lifeguard-perceived rip-current hazard $RHl$ was obtained for $\gamma = 0.23$, $z_{bar} = -3$ m and $d = 6.5$ m. Figure 5a shows the corresponding $V$ against $RHl$. It shows that $RHl$ increases with increasing $V$ and that for all hazard levels the values are nearly normally distributed, except for $RHm = 0$, which is clearly biased towards $V = 0$. The corresponding confusion matrix (Figure 5b) indicates that over the 558 hourly lifeguard-perceived hazard observations, 308 are correctly classified by the model. In line with the quantile approach used, the confusion matrix is symmetric, with a resulting accuracy of 0.55. However, by merging $RHm = 0, 1$ into low-hazard and $RHm = 2, 3, 4$ into moderate- to high-hazard hours (Figure 5c), the accuracy increases to 0.84, with a F-Score of 0.82, meaning that the model accurately predicts moderate to high rip-current hazard hours.

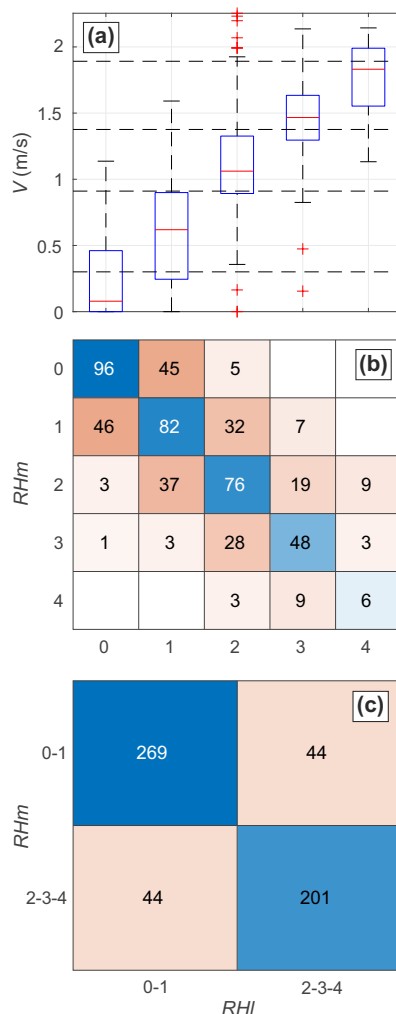

**Figure 5.** (a) Box plot of the modelled hourly rip-flow speed $V$ versus hourly lifeguard-perceived rip-current hazard $RHl$ on a 5-level scale. The central horizontal red marks indicate the median, the bottom and top edges of the box indicate the 25th and 75th percentiles, respectively, the whisker length indicates 1.5 times the interquantile range, and the crosses are the outliers. The horizontal dashed lines represent the limits between each hazard-perceived scale using a quantile-quantile approach. Corresponding confusion matrix of (b) hourly modelled ($SHm$) and lifeguard-perceived ($RHl$) rip-current hazard on the 5-level scale and (c) further discriminating low ($RH$=0,1) and moderate to high ($RH$=2,3,4) rip-current hazard hours.

Figure 6 shows the time series of wave and tide conditions as well as of the modelled rip-flow speed $V$, hourly modelled rip-current hazard level $RHm$ and hourly lifeguard-perceived rip-current hazard level $RHl$. Results show that $V$ is strongly modulated by tidal elevation $\eta$, with increased rip-current hazard for lower tidal elevations. On longer timescales, modelled rip current hazard is also modulated by the incident wave energy, with modelled hazard increasing with increasing wave height and wave period. Figure 6d,f,h further zooms onto a moderate-energy, average tide range, 5-day window showing that the tidal

modulation of the lifeguard-perceived rip-current hazard is very well captured by the model (Figure 6h). During a five-day period comprising the onset of a high-energy wave event with $H_s$ exceeding 2 m, the model also well captures the rip-current hazard which is maximised throughout August 20 (Figure 6e,g,h), which was also the only day of the summer of 2022 when the red flag was hoisted at La Lette Blanche beach, with lifeguard-perceived rip-current hazard maximized throughout the day.

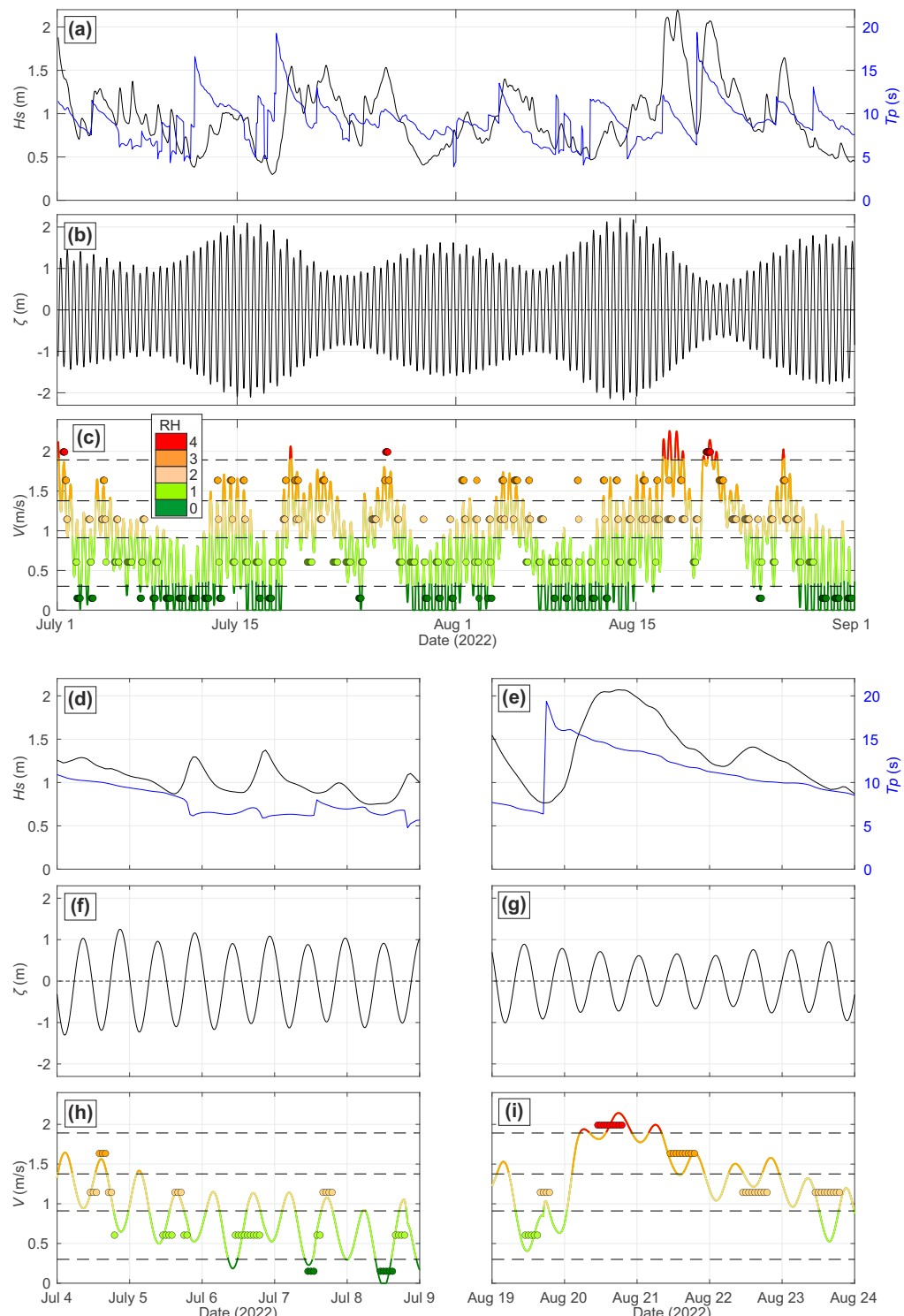

**Figure 6.** Time series of (a) significant wave height in 10-m depth $H_{s0}$ and peak wave period $T_p$; (b) astronomical tide level $\zeta$; (c) modelled rip-current flow speed $V$ and its corresponding hazard level $RHm$ (coloured line) and lifeguard-perceived rip-current hazard $RHl$ (coloured circles), which are further zoomed onto a 5-day period of (d,f,h) moderate-energy waves and average tide range and (e,g,i) high-energy waves and neap tides.

Figure 7 also shows that the model accurately predicts daily mean rip-current hazards. The pearson correlation between the daily-mean modelled $RHm$ and lifeguard-perceived $RHl$ hazards reaches $R = 0.83$. The model well captures most of the high-hazard days, although some are overestimated (e.g. August 17) or underestimated (July 28). The model also tends to slightly overestimate the daily-mean hazard during days when lifeguards perceived no rip-current hazard throughout the patrolling hours.

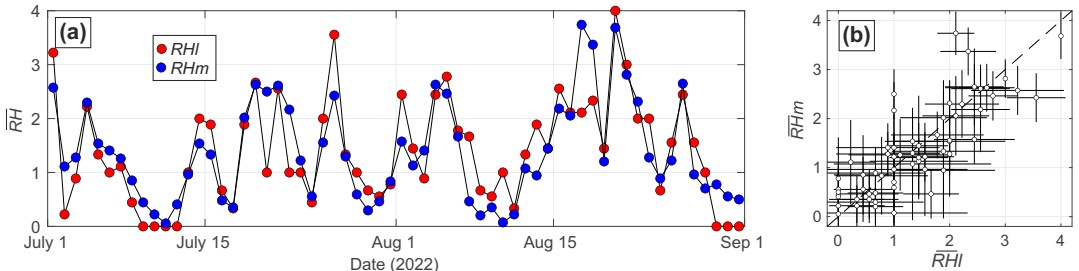

**Figure 7.** (a) Time series of daily-mean modelled (blue, $\overline{RHm}$) and lifeguard-perceived (red, $\overline{RHl}$) rip-current hazard on the 5-level scale. (b) Corresponding plot of $\overline{RHm}$ versus $\overline{RHl}$ with the horizontal and vertical lines indicating their daily standard deviation.

## 4.2 Shore-break wave hazard modelling

The best pearson correlation ($R = 0.70$) between the modelled shore-break wave energy $E_{sb}$ and the hourly lifeguard-perceived shore-break wave hazard $SHl$ was obtained for $a = -2.75$, $b = 0.3$, $\gamma_s = 0.4$ and $z_{bar} = -2$ m. Figure 8a shows that $SHl$ increases with increasing $E_{sb}$ with, for all hazard levels, values nearly normally distributed. In contrast with rip-current hazard results, which showed a limited number of outliers (Figure 5a), outliers are found for all lifeguard-perceived shore-break wave hazard levels $SHl$, except for $SHl = 1$. The corresponding confusion matrix (Figure 5b) shows that over the 558 hourly lifeguard-perceived hazard observations, 302 are correctly classified by the model, resulting in an accuracy of 0.54. However, and in line with what was found for rip-current hazard, by merging $SHm = 0, 1$ into low-hazard and $SHm = 2, 3, 4$ into moderate- to high-hazard hours (Figure 5c), the accuracy increases to 0.83 with a F-Score of 0.73, meaning that the model skilfully predicts the hours with moderate to high shore-wave break hazard.

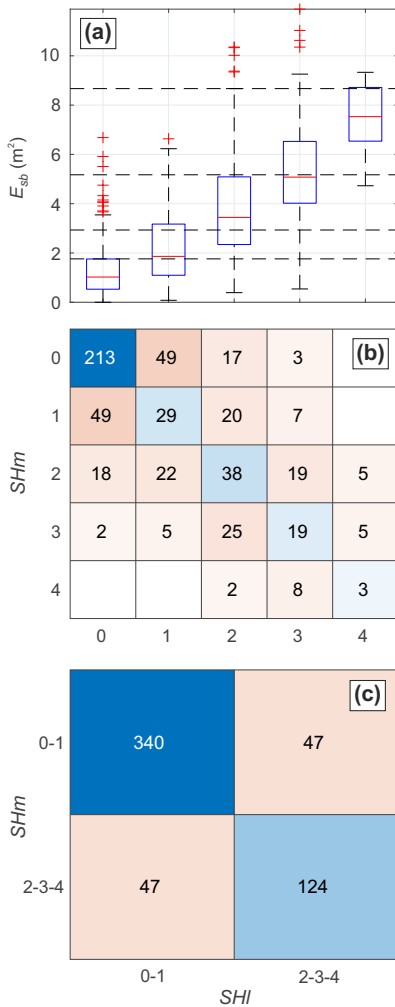

**Figure 8.** (a) Box plot of the hourly modelled shore-break wave energy $E_{sb}$ versus hourly lifeguard-perceived shore-break wave hazard $SHl$ on a 5-level scale. The central horizontal red marks indicate the median, the bottom and top edges of the box indicate the 25th and 75th percentiles, respectively, the whisker length indicates 1.5 times the interquantile range and the crosses are the outliers. The horizontal dashed lines represent the limits between each hazard-perceived scale using a quantile-quantile approach. Corresponding confusion matrix of (b) hourly modelled ($SHm$) and lifeguard-perceived ($SHl$) shore-break wave hazard on the 5-level scale and (c) further discriminating low ($SH$=0,1) and moderate to high ($SH$=2,3,4) shore-break wave hazard hours.

Figure 9 shows the time series of wave and tide conditions, as well as of the modelled shore-break wave energy $E_{sb}$ and
hazard level $SHm$, and hourly lifeguard-perceived shore-break wave hazard $SHl$. Results show that $E_{sb}$ is strongly modulated by the tidal elevation $\zeta$ with, in contrast with rip-current hazard, shore-break wave hazard maximized during the higher stage of the tide. In line with rip-current hazard, on longer timescales shore-break wave hazard increases with increased incident wave energy. Figure 9d,f,h further zooms onto a moderate-energy 5-day period moving from moderate to spring tides, showing

that the tidal modulation of the lifeguard-perceived shore-break wave hazard $SHl$ and the increased hazard with increased tide range are well captured by the model (Figure 9h). During a five-day period comprising the progressive decay in incident wave energy during nearly steady neap-moderate tides, the model also well captures the progressive decrease of shore-break wave hazard at high tides.

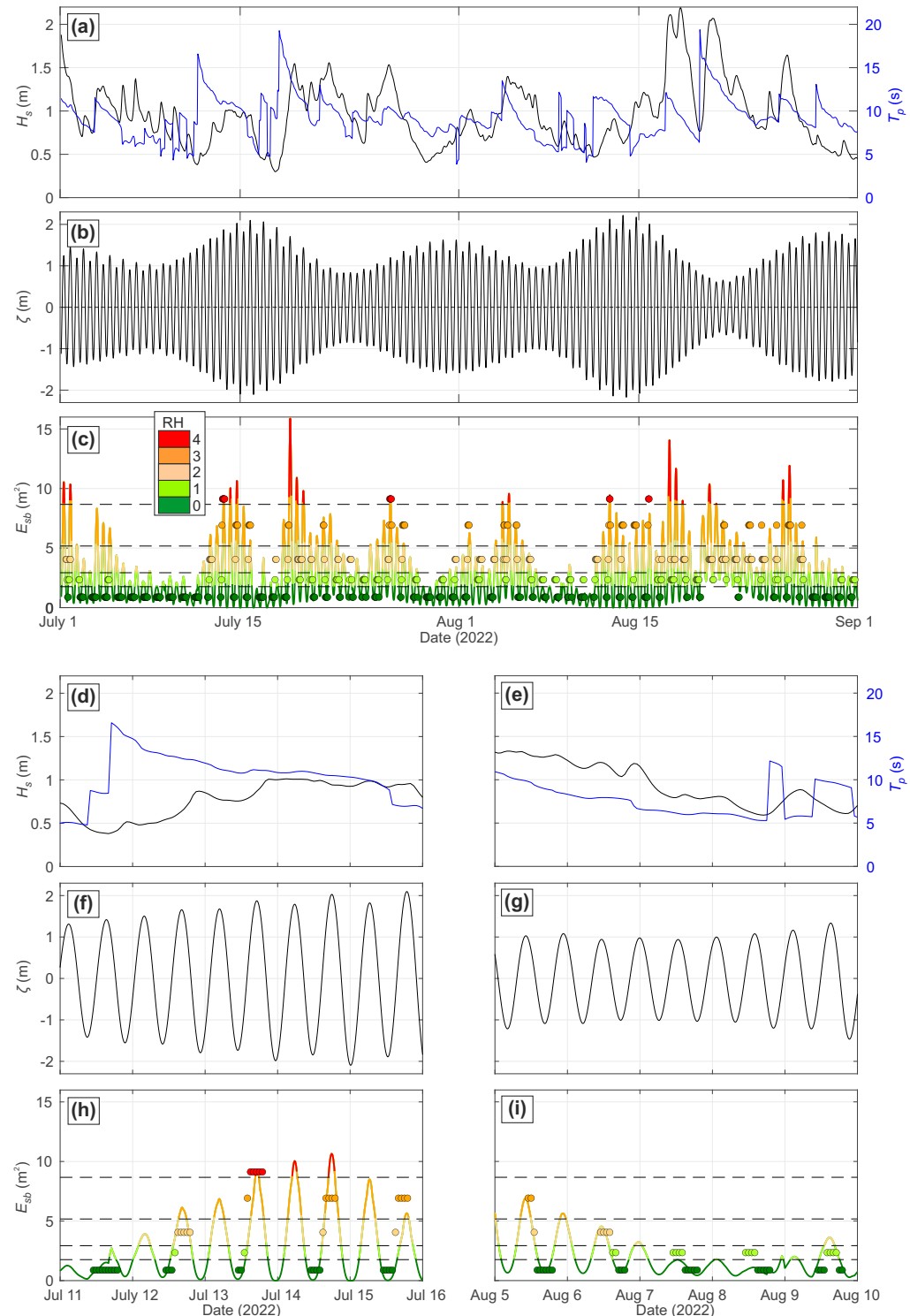

**Figure 9.** Time series of (a) significant wave height in 10-m depth $H_{s0}$ and peak wave period $T_p$; (b) astronomical tide level $\zeta$; (c) modelled shore-break wave energy $E_{sb}$ and its corresponding hazard level (coloured and circles), which are further zoomed onto a 5-day period of (d,f,h) moderate-energy waves and spring tides and (e,g,i) decreasing-energy waves and neap tides.

Figure 10 also shows that, although the largest lifeguard-perceived hazard days are underestimated by the model, the model fairly well predicts daily-mean shore-break wave hazards. The pearson correlation $r$ between daily-mean modelled shore-break wave hazard $SHm$ and daily-mean lifeguard-perceived shore-break wave hazard $SHl$ is 0.71. The model well captures most of the high-hazard days, although the two days with the highest lifeguard-perceived shore-break wave hazard (July 13 and July 26) are slightly underestimated by the model.

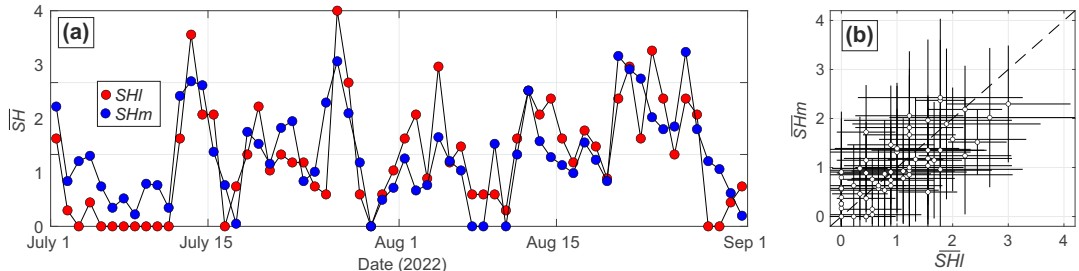

**Figure 10.** (a) Time series of daily-mean modelled (blue, $\overline{SH}m$) and lifeguard-perceived (red, $\overline{SH}l$) shore-break wave hazard on the 5-level scale. (b) Corresponding plot of $\overline{SH}m$ versus $\overline{SH}l$ with the horizontal and vertical lines indicating their daily standard deviation.

## 5 Discussion

Two simple semi-empirical rip current and shore-break wave hazard models were developed and further calibrated and tested on a high-energy meso- macro-tidal beach where the two surf-zone hazards co-exist. Previous beach hazard predictors essentially focused on rip currents with, to the best of our knowledge, only Casper et al. (2024) proposing a physics-based formulation. Their approach, based on the pioneering work of Moulton et al. (2017b), is consistent with the rip-flow model proposed here as it considers an idealised bar-rip morphology and the alongshore gradient in breaking-wave-driven-setup as the driving mechanism for rip current flow. In contrast with Moulton et al. (2017b) and Casper et al. (2024) our rip-flow model (1) is more simple as it does not discriminate between different surf zone conditions (shore-break, bar-break or saturated in Moulton et al., 2017b) as we consider a simple physics-informed random wave height decay law and (2) has a smaller number of free parameters. Given that our model appears to overestimate rip flow speed, comparison with field data should be performed in the future for calibration purpose. A detailed comparison between our model and that of Moulton et al. (2017b) and Casper et al. (2024) should be performed on different beaches to address for which type of morphological, tide and wave conditions a model performs better or worse. In addition, here based on lifeguard-perceived hazard on a 5-level scale, modelled rip flow speed $V$ was transformed into a similar hazard scale, showing very good skill (accuracy and F-Score exceeding 0.8) to predict moderate to high hazard hours ($RHl = 2, 3, 4$). The computed accuracy and F-Score are very good, and trying to further improve these metrics by complexifying the model may not be relevant. Indeed, as beach safety professionals, lifeguards are supposed to develop a more robust hazard perception than laypersons (Sandman et al., 1987; Slovic, 1999). However, according to Rowe and Wright (2001), it can also be argued that lifeguards remain human beings whose hazard perception can be influenced by personal factors (experience, gender, etc.). Using average lifeguard-perceived hazard data from all the

lifeguards on duty, instead of the chief lifeguard only, could provide a more robust data to calibrate the model. The validation approach proposed here can be applied anywhere pending lifeguard hazard assessment can be performed. If such lifeguard data cannot be collected, a first-pass approach is to base the hazard level scales on the threshold values computed in southwest

France (Table 1). Once again, such model application together with lifeguard-perceived hazard should be tested elsewhere to address the influence of beach state, modal wave climate and lifeguard perception on these threshold values. Since collecting consistent hourly lifeguard-perceived hazard data over a few weeks and under varying tide and wave conditions may not be feasible at many locations, an alternative approach is to use lifeguard-reported incidents (see, for instance, Scott et al., 2014). While such data also incorporate the exposure component of risk (Stokes et al., 2017), they are more widely available and can

be highly valuable, particularly for assessing whether the model can identify mass-rescue days.

Given that shore-break waves cause a large proportion of SZIs in southwest France (Castelle et al., 2018; de Korte et al., 2021; Castelle et al., 2024), we also proposed a shore-break wave hazard forecast model following a similar physics-informed approach. Combined, these two surf-zone hazard forecasts can provide detailed insight into surf-zone hazard evolution. This is illustrated in Figure 11 which shows the time series of rip-current velocity $V$ and hazard $RHm$, and of shore-break wave energy

$E_{sb}$ and hazard $SHm$ (Figure 11c,d), for an idealised time series of wave (Figure 11a) and tide (Figure 11b) conditions. For instance, this synthetic test case shows that rip-current flow and shore-break wave energy exhibit an out-of-phase behaviour with, under high-energy conditions (days 3 and 4), high rip-flow velocities sustained throughout the day ($V > 1.4$ m/s). In contrast, even under high-energy waves, shore-break wave energy only peaks during the highest stage of the tide. Therefore, if rip-current hazard gradually increases with increasing wave energy, still with higher hazard for low tide levels, shore-break

wave hazard is more modulated by tide, with shore-break wave hazard systematically absent at the lowest stage of the tide, even under high-energy waves (day 3 in Figure 11d).

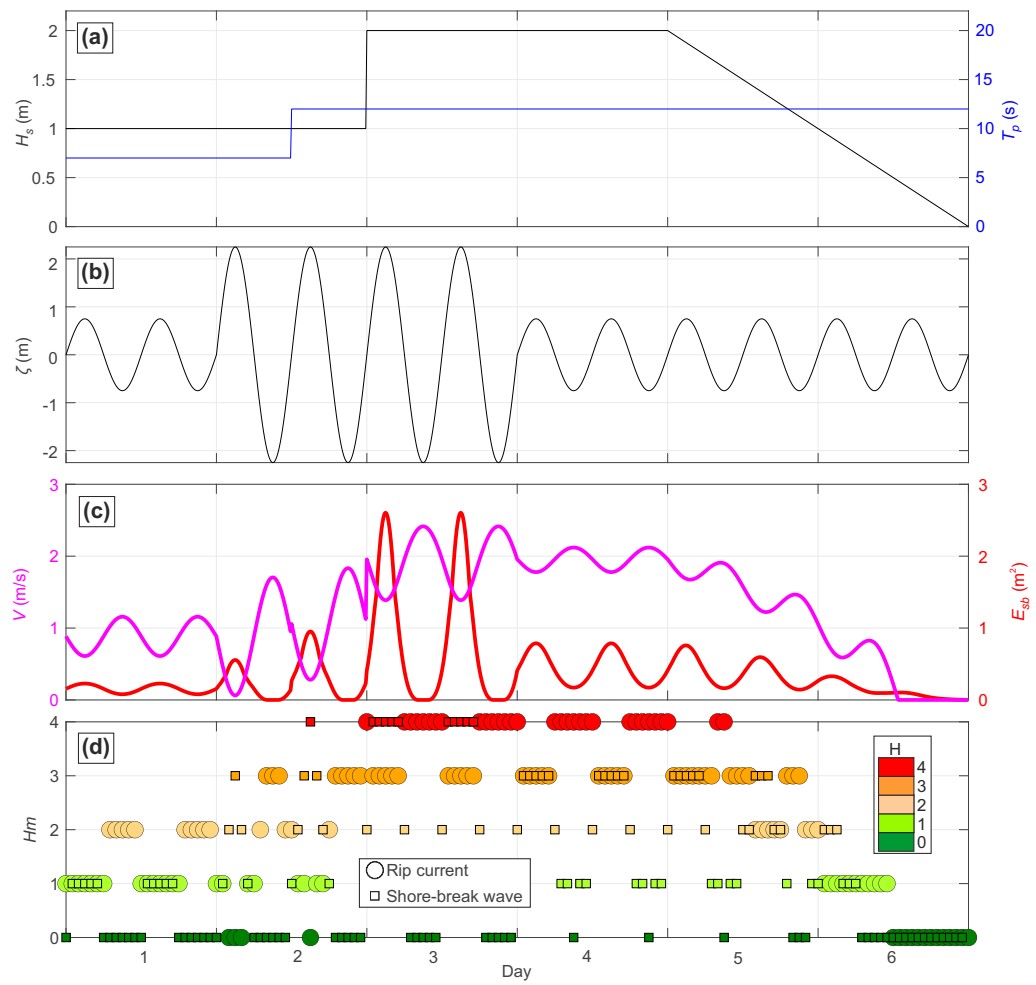

**Figure 11.** Synthetic time series of (a) offshore significant wave height $H_{s0}$ and peak wave period $T_p$; (b) astronmical tide level $\eta$; (c) modelled rip-flow speed $V$ and shore-break wave energy $E_{sb}$ and (d) their hazard level (coloured).

In addition to e.g. the 10-min or hourly rip-current and shore-break wave hazard forecasts, daily-mean hazard levels also showed very good skill, with a pearson correlation with daily-mean lifeguard-perceived hazards of $R = 0.83$ and $0.71$ for rip currents and shore-break waves, respectively. However, daily-mean hazard can be predicted with even simpler approach 320 i.e. based on the wave factor defined as $W_f = H_sT_p/|H_sT_p|$ (with the $|.|$ notation indicating the summer mean). By using this number introduced by Scott et al. (2014) to address rip-current rescues in UK, Castelle et al. (2019, 2024) showed that days with large $W_f$ values were associated a disproportionate amount of both rip-current related drowning and shore-break wave related injuries. During the summer 2022, the correlation between daily-mean $W_f$ and daily-mean lifeguard-perceived hazards reaches $R = 0.91$, which outperforms $\overline{RH}m$ ($R = 0.82$). Such improvement is not found with shore-break waves, 325 mostly because daily-mean shore-break wave hazard is much more affected by tidal range than daily-mean rip current hazard (Castelle et al., 2019, 2024). The daily-mean rip-current hazard forecast is important for providing a straightforward message

to the general public, and can also assist lifeguard managers in scheduling lifeguards in advance, ensuring they are deployed to the beaches where they will be most needed. In this context, the daily-mean wave factor ($W_f$) appears to be a simple yet powerful tool for predicting and communicating high rip-current hazard days. It is also important to note that the correlation

between the hourly lifeguard-perceived rip current hazard ($RH_l$) and the hourly wave factor ($W_f$) remains relatively high ($R = 0.65$). This indicates that, although $W_f$ alone does not account for tidal modulation, it still explains more than 40% of the observed variability in lifeguard-perceived rip current hazard. Overall, predicting daily-mean $W_f$ is complementary to the higher-frequency rip-current hazard hourly prediction throughout the day with our semi-empirical model, and to the shore-break hazard model which can be used for both daily-mean and hourly predictions.

Instead of using field data, here the models were calibrated based on lifeguard-perceived surf-zone hazard levels. The primary reason was that the bathymetry of La Lette Blanche beach was not surveyed during the summer of 2022, limiting the ability to estimate the bar/rip morphology metrics used in the rip-flow model. Instead, these metrics were found by maximizing the correlation between modelled rip-flow speed $V$ and lifeguard-perceived rip-current hazard $RHl$. The best model skill was found for bar crest elevation $z_{bar} = -3$ m and channel depth $d = 6.5$ m. These numbers are in line with previous detailed

surveys of some bar/rip morphology in southwest France (Sénéchal et al., 2011; Castelle et al., 2018). It must be noted that, while modelled rip-flow velocity is quite sensitive to the choice of the model free parameters, good skill is also found when using values significantly different from the optimal ones. For instance, the correlation between $V$ and $RH_l$ decreased slightly from 0.77 to 0.75 ($\approx -3\%$) when assuming a higher bar crest ($z_{bar} = -2$ m instead of -3 m) or a much shallower channel ($d = 2$ m instead of 6.5 m), which are closer to average values in southwest France. This suggests that a decent model skill can be

achieved with a rough estimate of the bar/rip morphology, further implying that temporal variability in beach morphology can be neglected in the model. Similar confusion matrix accuracy were also obtained as the thresholds (Table 1) are modified based the quantile-quantile approach. The optimal $H_s/h$ breaker indices ($\gamma = 0.23$, $\gamma_s = 0.4$) for random waves, sometimes referred to as the incipient breaker index, are different from the typical empirical breaker index (equivalent to $H/h$, with $H$ the individual wave height) used, for instance, in the parametric random wave models, which typically range from 0.6 to 0.8. In

line with previous field work (e.g. Raubenheimer et al., 1996; Power et al., 2010), our $H_s/h$ breaker indices for random waves are significantly smaller than 0.6-0.8.

    For the sake of consistency the free morphological parameters of the shore-break wave model were also found by maximizing the pearson correlation between shore-break wave energy $E_{sb}$ and lifeguard-perceived shore-break wave hazard $SHl$. When compared to the alongshore-averaged beach topography measured on July 12, 2022 at La Lette Blanche (Figure 12), the

355 Dean-like profile (solid blued line in Figure 12b) is much steeper than the alongshore-averaged profile. However, by changing $a$=-2.75 into $a$=-1.75, which is in much better agreement with the measured profile (dotted blue line in Figure 12b), the correlation between $E_{sb}$ and $SHl$ is approximately the same ($R$=0.70, with a marginal decrease by $\approx -0.5\%$ using the dotted blue line profile in Figure 12b). This once again shows that beach surveys can be used instead of a Dean-like profile calibrated with lifeguard-perceived hazards. The shore-break wave model was more sensitive to the shore-break wave height $H_{sb}$, i.e.

to the terrace/sandbar elevation $z_{bar}$ and breaker index $\gamma_s$. Overall, both the rip-current and shore-break wave hazard forecast models can be used based on some knowledge of the beach morphology. However, while parameters such as bar crest depth

and channel depth are relatively simple, obtaining them remains challenging due to the difficulty of surveying the surf zone, which is not routinely monitored at most locations. This raises important considerations for the large-scale transferability of the models. Future applications will need to determine how these parameters can be feasibly obtained, whether through direct

surveying, remote sensing, or empirical estimations based on regional morphology. Additionally, while the calibrated values used in this study may serve as a reference, their applicability to other sites remains uncertain, and further research is needed to assess whether re-calibration against lifeguard observations or other validation datasets is necessary at each new location.

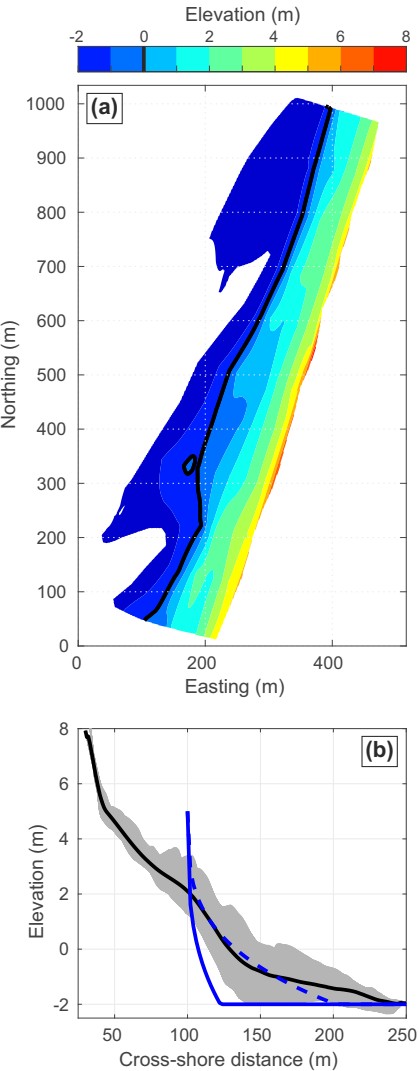

**Figure 12.** La Lette Blanche beach topographic survey performed at low tide on July 12, 2022 with (a) digital elevation model with elevation with respect to mean sea level coloured and (b) all cross-shore (light grey) and alongshore-averaged (thick black) profiles, with the solid (dashed) blue lines depicting the Dean-like profile for $a = -2.75$ and $b = 0.3$ ($a = -1.75$ and $b = 0.3$).

In line with (Moulton et al., 2017b), the rip-flow model does not seem to consider the wave period $T_p$. However, rip-flow speed is known to increase with increasing wave period, and we also show that wave period is key to the wave factor $W_f$ which outperforms daily-mean $V$ in explaining $\overline{RHl}$ variance. Surprisingly enough, including $(T_p/|T_p|)^n$ in Equation (5) did not increase model skill (best correlation was obtained for $n = 0$). However, $T_p$ is indirectly considered here in our model as the shoaled significant wave height $H_s$ is considered through the formulation of Larson et al. (2010), resulting in larger breaking wave height for larger period. This is why in Figure 11c the rip flow speed increases by 7% (from 1.71 to 1.83 m/s at spring low tide) during day 2 as $T_p$ increases from 7 to 12 s. Replacing the shoaled significant wave height at breaking $H_s$ by the significant wave height at the Météo-France wave model in nearly 10-m depth $H_{s0}$ slightly decreased the correlation between $V$ and $RHl$ from 0.77 to 0.76, showing the weak but positive influence of wave period $T_p$ on rip flow speed. In addition, under increasingly obliquely-incident waves, rip currents tend to progressively change from a symmetric seaward-flowing jet to an undulating longshore current (MacMahan et al., 2010). The influence of the presence of a longshore current component on the decay of rip flow was tested using the same approach as in Moulton et al. (2017a) and Casper et al. (2024). However, inclusion of the longshore current did not improve model skill. This is in agreement with Moulton et al. (2017b) who suggested that for deeper rip channels, like along the southwest France open beaches, rip-flow speed is not suppressed under obliquely incident waves. Including the effect of longshore current on rip flow speed is, however, strongly encouraged if applying the model on beaches with shallow rip channels (MacMahan et al., 2008). It must also be acknowledged that the rip current hazard in this study was estimated based solely on rip-flow speed. However, other flow characteristics can also influence the physical hazard, such as the rip current circulation regime, which plays an important role in determining the optimal rip-current escape strategy (McCarroll et al., 2014a). Surf-zone rip currents have long been perceived as narrow flows extending well beyond the breakers, rapidly flushing water out of the surf zone in what is known as the 'exit flow' circulation regime. However, studies using Lagrangian drifter measurements to compute surf-zone exit rates (e.g. MacMahan et al., 2010; McCarroll et al., 2014b) have shown that rip-flow patterns can also form quasi-steady, semi-enclosed vortices that retain most floating material within the surf zone, referred to as the 'circulatory flow' circulation regime. Unlike the exit-flow regime, the circulatory regime increases the likelihood that a swimmer caught in a rip current will be carried back to shallower, safer waters within a few minutes (McCarroll et al., 2015; Castelle et al., 2016a). Although observed and modelled exit rates in channel rips show considerable natural variability, the highest exit rates are generally associated with the lowest incident wave energy, and consequently, the lowest rip-flow speeds (see review in Castelle et al., 2016b). This study focused on channel rip currents, the most common rip type on intermediate beaches, although other types of rip currents exist (see Dalrymple et al., 2011; Castelle et al., 2016b; Houser et al., 2020). With the notable exception of Casper et al. (2024), who explored the potential for forecasting flash rip hazards at a Californian beach, hazard forecasting for other rip current types has never been tested. Our model is therefore mostly adapted for intermediate, high-energy, sandy beaches

The predicted natural hazard level is critical to communicate towards the general public as, by definition, it provides direct information on the level of threat of a naturally occurring event, here intense rip currents and powerful shore-break waves. However, the number of rescues and SZIs, which is of strong interest for lifeguard institutions and emergency units as it is the proxy of the volume of activity and thus of man power needs, also depends on the number of people exposing themselves

to the physical hazards (the exposure component described in Stokes et al., 2017). An option to predict beach risk is to fit a logistic regression model with SZIs data based on wave, tide and weather forecasts (Tellier et al., 2022). However, model skill strongly depends on the SZI dataset size and quality, and such models fail to identify the respective contributions of exposure and hazard components to the overall risk. The exposure component can be addressed through beach attendance, which can be computed with different techniques using e.g. video systems (Boominathan et al., 2016; Guillén et al., 2008). Given that beach attendance is largely governed by weather conditions (e.g. Dwight et al., 2007; Moreno et al., 2008; Ibarra, 2011; Coombes et al., 2009), as well as weekday and holiday periods (Kane et al., 2021; Tellier et al., 2022), machine learning techniques (e.g. Mahesh, 2020; Domingo, 2021) can be used to predict beach crowds. In order to robustly link up beach crowds and the number of people entering the water, which is exposure, the bathing rate will need to be addressed. Dwight et al. (2007) have estimated that, on average, only 45% of individuals arriving at the beach have physical contact with water on the southern California beaches. Such a proportion decreases during the colder winter months (26%) and increases in summer during warmer days (54%). Wave conditions can also influence the rate of bathing. For instance, de Korte et al. (2021) found that large shore-break waves ($H_s > 2.5$ m) can deter beachgoers from entering the water. Similarly, Dehez et al. (2024) demonstrated that weather and ocean conditions significantly impact beachgoers' risk perception and, consequently, their likelihood of entering the water. While further research is needed to improve predictions of exposure, the present work already provides valuable forecasts of the underlying hazard level. Since hazard itself is the primary concern for both the public and lifeguard services, these predictions can be highly useful even without explicitly accounting for exposure.

## 6  Conclusions

This paper introduces two new, simple, semi-empirical rip-current and shore-break wave hazard forecast models. These models, which depend on a limited number of free parameters, allow to estimate the time evolution of the rip current flow speed $V$ and shore-break wave energy $E_{sb}$. Using hourly lifeguard-perceived hazards collected over a two-month period, a quantile-quantile approach was used to transform $V$ and $E_{sb}$ into 5-level scale from 0 (no hazard) to 4 (hazard maximized). The forecast models accurately predict rip-current and shore-break wave hazard levels, including their modulation by tide elevation and incident wave conditions, opening new perspectives to forecast multiple surf-zone hazards on sandy beaches. The approach presented requires only a few beach morphology metrics, enabling surf-zone hazard prediction on beaches with wave forecasts. Combined with global beach safety research, this effort supports the development and communication of surf-zone hazard forecasts to help reduce drownings and surf-zone injuries.

*Data availability.*  The wave, tide and lifeguard-perceived rip-current and shore-break wave hazard datasets are available at https://osf.io/tzqax/

*Author contributions.* Conceptualization, data curation, formal analysis, model development, visualization and writing of the original draft was by BC; JPS collected lifeguard data; SL produced the wave data; Model validation by BC with guidance of DC; All authors reviewed and edited the draft; Project administration and funding by BC and JD.

*Competing interests.* The authors declare that they have no conflict of interest

*Acknowledgements.* This study received financial support from Project SWYM (Surf zone hazards, recreational beach use and Water safetY Management in a changing climate) funded by Région Nouvelle-Aquitaine, the French government in the framework of the University of Bordeaux's IdEx "Investments for the Future" program/RRI Tackling Global Change, as well as project IRICOT (PEPR IRIMA) managed by Agence National de la Recherche (ANR), France 2030, ANR-22-EXIR-0004, and Région Nouvelle-Aquitaine (PSGAR CORALi). We warmly thank the SMGBL (Syndicat Mixte de Gestion des Baignades Landaise), and particularly Stéphanie Barneix and the La Lette
Blanche lifeguards who were on duty during the summer of 2022. We are also thankful to the Vieille Saint-Girons council for providing technical support and access to the lifeguard facilities. The authors warmly thank the two anonymous reviewers for their insightful comments and constructive criticism which helped us to strengthen the paper.

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
