# Peer review of "Semi-empirical forecast modelling of rip-current and shore-break wave hazards"

_Natural Hazards and Earth System Sciences, 2024_

## Referee Comment (RC2)

Overview:

This is a well-written paper that is a significant contribution to forecasting rip-current and shore-break hazards using simple models informed by physics and calibrated with lifeguard observations of hazard levels. The introduction is thorough and logically organized and helpful physical parameter schematics are provided. Visualizations throughout are high quality. Time series of physics-informed parameterizations show remarkable agreement with lifeguard assessments of hazard. I found the idealized analysis showing how the models can be applied to hypothetical conditions to be interesting and informative.

  Prior to publication, I think the paper needs to provide more clear derivations and justification of assumptions leading to the new physics-based models; these could appear concisely in the main text or in a more detailed form in supplementary materials. There may be some errors in the rip-current speed and shore-break energy formulas, but it is difficult to assess without seeing more detail in how the authors reached those results. The rip-current hazard formulations based on rip-current speed have previously been derived and compared with lifeguard observations, and the authors derive their result from momentum balances (though more justification is needed). In contrast, the authors note that no theoretical estimate for shore-break hazard yet exists. The proposed shore-break formulation – the product of the Irribarren number and the wave energy – seems highly valuable, but given that it is somewhat ad hoc, maybe it would be more accurately described as semi-empirical or physics-informed rather than physics-based.

  The second broader comment I have is that some additional discussion of the limitations of this approach and its applicability to other sites would be helpful. Specifically, this approach seems to apply to sites where channel rips dominate, and the importance of other rip current types should be discussed. In addition, for applicability to other sites, it would be good to discuss how a minimal set of sandbar and beach profile shape parameters could be observed directly or estimated through tuning/calibration with lifeguard data, so that readers can assess feasibility.

  Line-by-line comments below indicate specific places where I suggest clarification on the physics-based parameterization and limitations/applicability.

Line-by-line comments:

- L36-38: "The most common rip type" - Clarify, this may be true on some beaches but not others

- L73: "The proposed framework offers new opportunities for forecasting rip-current and shore-break wave hazards at surf beaches with available wave predictions" - Morphology information also is needed, and ideally lifeguard observations for calibration. Consider adding these factors to the sentence.

- L131: "Rip current hazard can be estimated through the rip flow speed." Discussion section should cover how flow patterns and other factors may also affect hazard.

- L141, L146: "S=0.16*Hs", "Sb=0.16*Delta-Hsb", "Sc=0.16*Delta-Hsc" Please clarify under what assumptions these approximations are reasonable to use, and what assumptions are involved to modify the approximation for shoreline setup (as a function of wave height) to estimate setup immediately onshore of the bar and channel (cross-shore change in wave

height)? Does this assume breaking in the channel as well as on the bar? My intuition would say that Sb-Sc would then be independent of the offshore wave height, but the squared wave heigh decay equation suggests otherwise (see next comment). How does this more simplified approximation compare with other formulations that include more parameters, e.g., Moulton et al. 2017 / Casper et al. 2024? A simpler formulation with fewer parameters is ideal for hazard prediction if it is clarified under what conditions it is a reasonable approximation. It seems like this formulation could be roughly a factor of 4 larger than Moulton/Casper, but I'm not completely sure, especially given the complexity of the quadratic delta-H formula.

- L148-151: "Here we consider simple first-pass estimation of the significant wave height decay for irregular waves." – Is there a reference for this? Or provide a derivation or more explanation. Assuming a wave breaking gamma and single wave height, I would expect Delta-Hs to be simply Hs-H, where H=gamma*h for broken waves. Does Equation (2) differ from this due to considering an irregular wavefield, e.g., Rayleigh distributed wave heights?

- L150: It could be worth spelling out the two equations for Hsb and Hsc, so that the dependence of the speed on the bar-channel geometry is clearer

- L152: Please provide references and/or justification for the simplified momentum balance

- L155: I think more justification is needed for these approximations. Is it known that the setup varies over a lengthscale of the width of the channel? Why not a half-width, or a multiple of the width, or something else like the spacing between channels, or a frictional lengthscale? I don't think this is actually known. Similarly, for the advective term, given the argument is that this is a physics-based parameterization, a derivation should be provided. Using the continuity equation with the left-hand side of Equation 3, it is not clear how the $2*V^2*h/w$ approximation is reached. Are assumptions made about U=V or U=1/2*V or U=2*V? Is the alongshore lengthscale w or ½*w or 2*w? Is it assumed that alongshore depth variations are small (dh/dx * 1/h is small)?

- L157: (Equation 4) I'm not convinced this formula is correct. The Moulton 2017 / Casper 2024 formula would be sqrt(2*g*(Sb-Sc)), which is different from this by a factor of 2. The Sb-Sc formula may have an extra factor of 4 relative to the Moulton 2017 setup difference estimate. Interestingly, these differences would compensate each other. I would have most confidence in a formulation that is consistent with past work that has been compared with field observations of speeds.

- Figure 3: The way S(x) is drawn as a square wave, dS/dx is not differentiable... would it make sense to show linear variations in S from the bar to the channel center instead?

- Figure 3d, 4b: I am confused by the diagrams in Figure 3d and 4b. What are the x and y axes?

- L165: "no theoretical framework to estimate a measure of the shore-break wave energy" – If this is the case, I might describe the following formulations as physics-informed rather than physics-based, but this is a wording nuance

- L186: Does the squared quantity come from the same "decay law" used in the rip-current formulation? Could write this as a 3-part equation for wave-breaking types (subaerial bar, bar-breaking, and shoreline-breaking)?

- L188: Could $Z\_l$ be written as z_bar, for consistency with the rip-current formula?

- L169: "deep water wavelength" - Is it possible that the wave condition upon shore-breaking deviates from the deep-water wavelength, since breaking on the bar could filter out some frequencies given differences in steepening and breaking? Particularly for wavefields with broad or multi-peaked frequency spectra. Could you comment on when using offshore wavelength is relevant?

- L169: Should this be Tp squared?

- L173: Is a factor of sqrt(2*pi) missing in the equation?

- L195: "thresholds were computed in order to obtain the same number of modelled hazard levels" – does it need to be exactly the same number? You could allow some uncertainty to avoid overfitting / specify confidence intervals on this choice of ranges. I doubt the confidence is reflected in the significant digits shown, with 1 cm/s and 0.01 m^2 resolution.

- L200: "daily-mean" – Is the mean, max, or median most relevant for hazard? I would think maximum may be most relevant. Daily is somewhat coarse. I wonder about having at least having morning and afternoon to capture some of the tidal variability, and could be relevant for shift staffing by lifeguards.

- L208-211: "by merging […] into low-hazard […and…] moderate- to high-hazard hours […], the accuracy increases" – It would be worth discussing here or in the Discussion why the 5-level scale did not perform well. Was it because there wasn't enough data or that the parameterizations capture a clear enough relationship between inputs and outputs to predict hazard on a finer scale?

- L230: "outliers" – Might these be worth discussing further since hazardous events that are "outliers" and not well forecast could be dangerous.

- Figure 6,9: Since panels a and b are duplicated in these two figures, consider merging these in one figure with both of the full the rip current and shore-break time series, which may be interesting to show how they vary differently with conditions (similar to Figure 11). The example shorter time window in panels d-i could be two separate figures for rip currents and shore-break. Just a suggestion.

- Figure 7,10: Would a bin average help to show if the model tends to be over- or under-forecasting at different hazard levels?

- L269: "should be tested elsewhere" – Here or in the Discussion (could go with paragraph beginning on line 295 in the Discussion), it would be good to discuss how the sandbar elevation and beach profile shape parameters can be inferred, and/or the need to get these morphology parameters through tuning/calibration with lifeguard data, which is also hard to get. In addition, note that this approach assumes that the beach is always channeled, and that channel rips are the strongest rips, as opposed to transient rip currents, structural rips, etc.

- L283: "daily-mean lifeguard perceived hazards" - Would daily max be better for hazard preparation, given that the mean could obscure a brief but high-risk time period? Or split into morning vs afternoon max or mean?

- Figure 12: Why is the Dean profile so different from the measured profile?

- L322: "weak but significant" - Is this statistically significant?

- L366: "only a few basic beach morphology metrics" - This may be a little vague and subjective use of "basic," clarify.

- L299-L305: d=6.5 m seems like an unrealistically deep channel. Did you consider constraining the parameter range in the fit to physically realistic values? The skill was similar for more realistic values so this would not change the results much but could provide more realistic predictions for future conditions.

Minor typographic suggestions:
- L2: change "can expose to" to "can be exposed"
- L5: change "allow to compute" to "can be used to compute the time"
- L8: August date missing, August 31?
- L12: change "where wave forecast is available" to "where wave forecasts are available"
- L14: remove "e.g."
- L17: remove "been" in "have been greatly increased"
- L18: remove "e.g."
- L43: lengthy paragraph- could start a new paragraph at "Shore-break"
- L69: change "estimate" to "estimates"
- L70: add "an" before "associated 5-level scale"
- L71: change "can be either given thanks to" to "either be given based on"
- L71: remove "e.g."
- L103: change "surf-zone hazard forecast" to "surf-zone hazard forecasts"
- L104: change "numerical wave hindcast instead" to "numerical wave hindcast data instead"
- L104: change "consisted in an analysis" or "consisted of an analysis"
- L108: change "as unified" to "including unified"
- L111: change "was assumed representative" to "was assumed to be representative"
- L126: here "RH_l" and "SH_l" have a "l" subscript- correct typos throughout the manuscript where "l" is not subscripted- same comment for "m" subscript
- L135: notation- consider using x as cross-shore coordinate and y as alongshore coordinate for consistency with most of the surfzone literature, also consider eta-bar for setup instead of S, which is typically used for radiation stress
- Figure 3: Hard to see the red text on the blue background in panel a.
- Figure 4: Consider changing the notation for the terrace elevation, Z_l, elsewhere "l" is used for lifeguard.
- L171: Should the subscript be ssb for H?
- Figure 5,8: is it typical for the y axis to be flipped like this in the confusion matrix?
- Figure 11: hard to see the difference between the dark pink and red, could switch to red-blue colors in panel d or other colorblind friendly palette
- L285: add "indicating" after "notation"
- Figure 12: consider switching to a colorblind friendly colorbar, caption unclear- suggest rephrasing to "with the blue solid (dotted) lines depicting…"
- L332-L334: remove unnecessary use of "e.g."
- L336: change "model" to "models"

---

## Author Comment (AC1)

**General comments**

\* This paper presents a novel way to predict rip current and shore break wave hazards, providing a new means to forecast these coastal hazards ahead of time. It uses a relatively novel physics-based approach to predicting these processes. The methods have been calibrated and validated over a single summer season at a beach in France using lifeguard perceived hazard estimates, indicating that the system performs very well against those test data. This contribution has the potential for wide reaching impacts in coastal hazard prevention through forecasting rip and shore break hazards (which is alluded to in the introduction), as long as the authors can address some of the issues I raise below regarding transferability of the approach.

> *Reply: We thank Reviewer #1 for their support for publication and constructive criticism. In our detailed response below you will see that all the comments have been carefully considered and required changes have been made. We warmly thank Reviewer #1 for their comments which really helped strengthening our paper.*

\* The paper addresses a relevant scientific question that is within the scope of the journal. The approach is novel for the purposes of hazard prediction and the methods are clearly outlined. The results are sufficient to support the conclusions reached. In the introduction you argue that previous approaches have been 'validated on a limited number of beaches' and that 'more generic surf-zone hazard models' are required 'to be applied to a wide range of sandy beaches'. Given this justification for developing a new method of rip forecasting, I think more discussion is warranted on how feasible the methods would be for a large number of beaches (e.g. a national-scale rip forecast) and how transferable the methods are for non-lifeguarded beaches. I.e., is this a method that's useful and accurate but is only really feasible for a handful of high-risk sites where it's worth undertaking a lengthy calibration effort to gather lifeguard perceptions, or is this a method that can realistically be generalised and tested on a regional or national-scale? If so, briefly indicate how you would propose for this to be done?

> *Reply : Reviewer #1 raises an important point, and we agree that how our approach can be applied and transfered at larger scale deserves more discussion. This is now addressed in the Discussion section of the revised manuscript (see also reply to last specific comment) : "However, while parameters such as bar crest depth and channel depth are relatively simple, obtaining them remains challenging due to the difficulty of surveying the surf zone, which is not routinely monitored at most locations. This raises important considerations for the large-scale transferability of the models. Future applications will need to determine how these parameters can be feasibly obtained, whether through direct surveying, remote sensing, or empirical estimations based on regional morphology. Additionally, while the calibrated values used in this study may serve as a reference, their applicability to other sites remains uncertain, and further research is needed to assess whether re-calibration against lifeguard observations or other validation datasets is necessary at each new location."*

**Specific comments**

\* Line 53: consider citing Stokes et al. (2024) 'New insights into combined surfzone, embayment, and estuarine bathing hazards' here for a current reference. They predominantly forecast estuary hazards, but they also forecast rips with a process-based model

*Reply : Thank you for this suggestion. Stokes et al. (2024), which was not published at the time we submitted our manuscript is now cited where listing the different rip current forecast approaches*

\* Line 57: You should also cite Scott et al. (2014), 'Controls on macrotidal rip current circulation and hazard' here, which was their earlier paper describing the data and analysis, while this reference describes the application to lifeguarding. I would also mention that this final approach relied on lifeguard incident data for calibration/validation of thresholds

*Rely : Thank you for pointing this reference which is now included with mention of the lifeguard incident data*

\* Lines 70-72: This could do with rewording - it's not been mentioned yet what the free parameters are, which in itself is not a problem, but it makes it confusing when you mention setting up the parameters with either beach morphology (presumably an input parameter) or lifeguard-perceived hazard data (presumably the target variable). Consider briefly summarising what the free parameters are, or explain how they relate to beach morphology and lifeguard perceived hazard.

*Reply : This is now specified "... only require a limited number of time-invariant free parameters related to beach morphology and wave breaking onset. These parameters can either be given based to some knowledge of the beach morphology, or through calibration using lifeguard-perceived hazard data."*

\* Lines 72-73: This may be dealt with later in the discussion, but you mention earlier that a shortcoming of previous systems is that they were calibrated on only a few beaches, and that more generic models are needed to apply to a wide range of sandy beaches. However, in this study you only use a single beach. I would suggest in this final paragraph of the introduction you should manage the readers expectations - is this new system demonstrated on a single beach, or is it shown to be applicable to a wide range of sandy beaches?

*Reply : We fully agree – We now specify that it is a single-beach application : "The proposed framework, here applied to a single beach in southwest France, offers new opportunities for ..."*

\* Lines 106-107: While the specifics of this Wavewatch model are not fundamental to your conclusions, it does influence your results to some extent, and I therefore think slightly more detail about the wave model is warranted. It would be good to know the extents of this model - is it a local area model or is it simulating waves along the entire French coast? How far offshore does it extend? Was it developed for this study? If this information is in another paper, then you can cite that instead.

*Reply : Thank you for this comment. There is little meterial published on this model, which is the operational model of Météo-France running for operational sea state forecasting. More detail is now given in the revised manuscript : "The MFWAM (Météo-France Wave Model) based on the spectral wave model WAM (WamdiGroup, 1988) is the French version of the European Centre for Medium-Range Weather Forecasts (ECMWF) WAM model used by Météo-France for operational sea state forecasting, with a 0.1◦ grid resolution in the northeast Atlantic. It forces a high-resolution WaveWatch 3 wave model (Tolman et al., 2002), forced by winds from the ARPEGE model of Météo-France. The model uses an unstructured grid (Roland and Ardhuin, 2014), allowing the French Atlantic coast to be described with a*

*resolution of approximately 200 m, with mesh size increasing to approximately 10 km at the boundary of the model a few hundred of kilometres offshore. Different coastal processes are represented in this model, such as unified parameterization of wave breaking from offshore to coast, wave reflection at the coast, refraction due to currents and bathymetry, and bottom friction. Modelled wave conditions were extracted in approximately 10-m depth in front of La Lette Blanche beach, i.e. to estimate the wave conditions outside of the surf zone. "*

* Line 106: I suggest re-wording the sentence slightly - Wavewatch can use an unstructured grid, but it doesn't have to use one. I would briefly mention the min and max grid resolutions used and explain that the unstructured grid allows computational efficiency. You could after all resolve the French coast at 200 m with a regular square grid (although you wouldn't necessarily want to!).

> *Reply : see reply ro previous comment*

* Line 110: As you've mentioned that the model can simulate coastal processes such as wave breaking, I think it's worth mentioning here explicitly that the model is not being used to simulate surfzone conditions - it is being used to estimate wave conditions just outside the surfzone.

> *Reply : see reply to previous comment*

* Line 112: Please state over what time period the results were gathered.

> *Reply : this is now specified : "Over the period from July 1 to August 31 of 2022, results show ..."*

* Line 134: I'm not sure I agree with how this is worded – rather than saying 'alongshore variability in depth of the sandbar' isn't it more accurate to say something like 'alongshore variability in depth between the sandbars and intervening drainage channels'

> *Reply : We agree with this rewording which can now be found in the reviesed manuscript.*

* Line 147: The Larson et al. (2010) equation is absolutely fine for the purpose it's being used, but you should give some mention to what the equation includes and excludes. As a minimum, you should mention that this is a simple linear shoaling equation with a refraction law included, and assumes a simple linear seabed slope. It doesn't consider complex (barred) surfzone slopes or bed friction. To be more complete, you could include the full formulation (which is arguably appropriate, given that it is a key equation in your predictive system).

> *Reply : Thank you for raising this point. The full formulation includes several equations that, in our view, would not enhance the readability of the paper. Instead, we provide a brief summary of the primary assumptions : "This formula allows to compute the incipient breaking wave properties based on a simplified solution of the wave energy flux conservation equation combined with Snell's law, assuming shore-parallel depth iso-contours."*

* Equation 2: it is not completely clear from this formula which Hs is being used here - i.e. is this Hs at breaking, or does this Hs need to be locally defined? Please clarify this in the preceeding or proceeding text.

> *Reply: Indeed this is Hs at breaking i.e. computed through the Larson formula, this is now clarified*

* Equation 4: Intuitively, I would assume the velocity in the rip channel will depend on the gradient in wave setup between the bar and the channel, not the absolute difference between the two.

However, I see from your diagram below how this can be neglected in an idealised case. As these gradients are present in equation 3, please explain in conceptual terms (and referencing the figure below) why the x and y dimensions can be safely neglected and are no longer important in order to estimate the flow velocity.

> *Reply : Thank you for this comment. We agree that such simplicification is not straightforward. This is now briefly explained in the revised manuscript, also referring to Moulton et al. (2017b) who developed this in more detail : "Following Moulton et al. (2017b), we assume that the ratio of bottom stress to the advection term is small, and that the balance of pressure gradients and advection along a streamline can be approximated using the Bernoulli equation. By further neglecting the effects of inertia in a longshore current driven by obliquely incident breaking waves, the rip flow velocity V can be approximated as:"*

* Line 171: Wave power at breaking can be derived from the wave breaker energy ($E_b = 1/8$ rho g $H_b^2$) and shallow water group velocity ($C_g = \sqrt{g\ hb}$) as $P_b = E_b C_g$. This would seem a more obvious way to compute breaker wave power. I assume your motivation to derive a different formula here is because the local beach slope is not considered in the Larson (2010) calculation of Hb, and that this therefore provides an inferior estimation of wave power if you happen to know (or can estimate) the local beach slope. Please explain and justify in the text why you choose this approach over a more common airy wave theory approach.

> *Reply : We chose to use a proxy of wave energy instead of the energy flux for the sake of simplicity. We, however, tested different parametrisation like, using the wave period (used in Cg or in the wave factor Wf) however, it did not improve the results.*

* Line 175: This equation is commonly presented as $h = Ay^{(2/3)}$. Please cite where this version comes from and define how you use this equation. For example, a is usually a sediment dependent scale parameter and b is usually taken as 2/3; How have you defined them in this application?

> *Reply : This is a modified Bruun/Dean profile, typically expressed as $h=Ax^m$, where A and m vary from site to site. Although m=2/3 is often assumed to be a good approximation, in this case, we account for large tidal variations and must also consider the subaerial beach. For this reason, we added 5 m so that z(x=0)=5. By not fixing A and m, we aimed to give the model as much flexibility as possible to determine the best-fit parameters, especially since we are dealing with an intertidal, potentially bermed profile. This differs significantly from the idealized Dean profile, which typically extends deeper. This clarification has been incorporated into the revised manuscript and we now do not refer to Dean profil to avoid confusion: "Note that here we did not consider a Dean profile (b=2/3) because be are interested in the intertidal, potentially bermed, part of the beach profile"*

* Line 184: 'in between, offshore wave breaking occurs' - I suggest using a different term here as 'offshore' sounds seaward of the surfzone/ sandbars. However, I assume you are referring here to wave breaking on the sandbar?

> *Reply : The Reviewer is right, we now clarify : "... there is no wave breaking across the terrace, if ..."*

* Line 191: The quantile-quantile approach you used to transform your values into a 5-level scale warrants explanation in the text. How exactly was this done?

*Reply : this is explained a couple of sentences later in the paragraph : "Second, the values of V and Esb concurrent to lifeguard observations were sorted and thresholds were computed in order to obtain the same number of modelled hazard levels (Table 1). Based on these ranges of V and Esb , the complete time series of V and Esb were transformed into modelled rip-current (RHm) and shore-break wave (SHm) hazard on the same 5-level scale as for lifeguard observations."*

\* Line 195: Please explain in more detail how thresholds were determined to distinguish the five levels.

*Reply : We are not sure to fully understand your concern as this is further explained later in the paragraph "the values of V and Esb corresponding to lifeguard observations were sorted, and thresholds were computed to ensure the same number of modeled hazard levels." This means, for example, that for the first threshold distinguishing between levels 0 and 1, if there were 25 lifeguard observations of a level 0 rip current hazard, the threshold for V was set at the 25th smallest value of V. This process was repeated for all hazard levels.*

\* Line 247: to be completely transparent, the top 4 lifeguard perceived hazard values are understimated by the model. However, the correlation and performance is generally very impressive.

*Reply : We agree, this sentence now reads "Figure 10 also shows that, although the largest lifeguard-perceived hazard days are underestimated by the model, the model fairly well predicts daily-mean shore-break wave hazards. "*

\* Discussion: Another point that may be interesting to investigate (although entirely optional) now that you have well calibrated models, is what proportion of time this beach exists at each hazard level. This would simply require running the models over a longer time frame (a few years of wave and tide data, for example) and plotting the distribution of different hazard levels for rips and shore break waves. As I say, this is an entirely optional suggestion.

*Reply : This is an extremely relevant comment! This analysis is planned as part of a broader study examining the impact of climate change on summer wave conditions and, in turn, surf-zone hazards. Specifically, we aim to explore how the proportion of each hazard level may evolve over the long term during summer. Although morphological changes will necessarily be excluded, this will provide a preliminary assessment of potential changes in surf-zone hazards.*

\* Line 266: Another approach that is probably worthy of discussion and that follows from previous papers (scott et al (2014), for example) would be to test the developed models against lifeguard recorded incident data. This would have benefits and limitations (e.g. mixing risk and hazard), but it would be interesting and valuable to see how well the models pick out periods of incident occurrence. This may be one of the only feasible ways to test the model's applicability on a large number of beaches, where gathering lifeguard perceptions may not be so feasible.

*Reply : We fully agree. This is now included in this paragraph of the discussion section : " Since collecting consistent hourly lifeguard-perceived hazard data over a few weeks and under varying tide and wave conditions may not be feasible at many locations, an alternative approach is to use lifeguard-reported incidents (see, for instance, Scott et al., 2014). While such data also incorporate the exposure component of risk (Stokes et al., 2017), they are more*

*widely available and can be highly valuable, particularly for assessing whether the model can identify mass-rescue days"*

\* Lines 266-267: 'The validation approach proposed here can be applied anywhere pending lifeguard hazard assessment can be performed' - This is not a trivial undertaking! Can you comment on how many lifeguard observations would be required at each site to robustly tune the models?

> *Reply : We fully agree that this is very challenging. At this stage it is hard to say, but it would require quite a few weeks of hourly lifaguard estimations for representative wave and tide conditions, this is why we precise (see also reply to previous comment) : " Since collecting consistent hourly lifeguard-perceived hazard data over a few weeks and under varying tide and wave conditions may not be feasible at many locations..."*

\* Lines 288-289: For completeness, can you comment on how Wf performs when applied hourly? Presumably, the model you present here performs better at hourly resolution as it captures tidal variation?

> *Reply: We warmly thank Reviewer #1 for this insightful suggestion. The reviewer is correct that our model performs better than Wf at an hourly resolution as it captures tidal variations. However, the correlation between hourly Wf and lifeguard-perceived rip current hazard remains quite high (R = 0.65), indicating that despite tide modulation, Wf = Hs\*T accounts for more than 40% of the observed lifeguard-perceived rip current hazard variability. This has now been included in the discussion section of the revised manuscript "It is also important to note that the correlation between the hourly lifeguard-perceived rip current hazard (RHl) and the hourly wave factor (Wf ) remains relatively high (R = 0.65). This indicates that, although Wf alone does not account for tidal modulation, it still explains more than 40% of the observed variability in lifeguard-perceived rip current hazard."*

\* Line 291: daily mean hazard would also help lifeguard managers roster lifeguards, some days ahead, to the beaches where they will be most needed

> *Reply : We, once again, fully agree, the text has been modified into : "TThe daily-mean rip-current hazard forecast is important for providing a straightforward message to the general public, and can also assist lifeguard managers in scheduling lifeguards in advance, ensuring they are deployed to the beaches where they will be most needed. In this context, the daily-mean wave factor Wf) appears to be a simple yet powerful tool for predicting and communicating high rip-current hazard days. "*

\* Line 304: This is an important point, as it suggests that the predictive method is not sensitive to the sort of changes in the bar and channel that might be expected within a single season. Can you comment on what a typical range of bar elevations and channel depths are expected to be (at least at this beach)?

> *Reply: Thank you for this comment, this is now clarified in the revised manuscript : "For instance, the correlation between V and RHl decreased slightly from 0.77 to 0.75 (≈ −3%) when assuming a higher bar crest (zbar = −2 m instead of -3 m) or a much shallower channel (d = 2 m instead of 6.5 m), which are closer to average values in southwest France. This suggests that a decent model skill can be achieved with a rough estimate of the bar/rip morphology, further implying that temporal variability in beach morphology can be neglected in the model."*

* Line 318: Please comment on the calibrated gamma value you found in this study - it is significantly lower than would be typically expected. How sensitive is the model to this value? What correlation would be obtained if you used a more typical value for gamma?

> *Reply: Thank you for this comment. Please note that the breaker index used here is for random waves, which differs from the breaker index for regular waves, which indeed typically ranges from 0.6 to 0.8. This has now been clarified in the revised manuscript : "TThe optimal Hs/h breaker indices (γ = 0.23, γs = 0.4) for random waves, sometimes referred to as the incipient breaker index, are different from the typical empirical breaker index (equivalent to H/h, with H the individual wave height) used, for instance, in the parametric random wave models, which typically range from 0.6 to 0.8. In line with previous field work (e.g. Raubenheimer et al., 1996; Power et al., 2010), our Hs/h breaker indices for random waves are significantly smaller than 0.6-0.8."*

* Line 348: This sounds like it's a limitation of your study, but that's only true if predicting overall risk is of interest (for lifeguard resourcing, for example). I suggest re-iterating here that even without predicting exposure the present system provides useful prediction of the underlying level of hazard, which is the primary factor of interest to both the public and lifeguard services.

> *Reply : Thank you for providing this insightful comment and indeed it read too much like a limitation of our work. Based on your comment this now reads : "While further research is needed to improve predictions of exposure, the present work already provides valuable forecasts of the underlying hazard level. Since hazard itself is the primary concern for both the public and lifeguard services, these predictions can be highly useful even without explicitly accounting for exposure."*

* Conclusions: As per my general comment regarding the discussion of transferability to other sites, I think the conclusions need to at least briefly address how feasible it is to apply the developed models at a large number of sites. i.e. how can these models feasibly be calibrated/validated at other locations? You need to be more realistic about how feasible it is to collect the required parameters at other sites. The rip model morphological parameters (bar crest depth and channel depth) are 'simple', but they are not trivial to measure. As the authors know, the surfzone is notoriously difficult to survey and is not routinely surveyed by monitoring programmes, and only at select locations globally is measured occasionally for research purposes. Therefore, the conclusions need to briefly address how these parameters are expected to be gathered for future application of these models, especially if they are to be useful for a large number of sites. Are the calibrated values used here expected to be applied elsewhere (along with some form of validation)? Are the parameters expected to be re-calibrated at each new location against lifeguard observations? or perhaps through direct surveying of the surfzone morphology?

> *Reply : Thank you for raising this important point. In line with this limitation we decided to add some text to the 4th paragraph of the discussion section, otherwize the conclusions section would have been too negative. This new text reads : "However, while parameters such as bar crest depth and channel depth are relatively simple, obtaining them remains challenging due to the difficulty of surveying the surf zone, which is not routinely monitored at most locations. This raises important considerations for the large-scale transferability of the models. Future applications will need to determine how these parameters can be feasibly obtained, whether through direct surveying, remote sensing, or empirical estimations based*

*on regional morphology. Additionally, while the calibrated values used in this study may serve as a reference, their applicability to other sites remains uncertain, and further research is needed to assess whether re-calibration against lifeguard observations or other validation datasets is necessary at each new location."*

**Technical corrections**

*Reply : All the technical corrections suggested below have been made, except where a specific reply is provided, and with a more specific reply for the comment on figures 3d and 4b.*

* Line 2: Change 'expose' to 'be exposed'

* Line 2: I suggest changing 'The most severe and widespread natural hazards' to ' The most severe and widespread natural bathing hazards'

* Line 8: Change 'from July 1 to August, 2022' to 'during July and August of 2022'

* Line 9: Change 'into' to 'into a'

* Line 12: Change 'where wave forecast is available' to 'where a wave and tide forecast are available'

* Line 39-40: Rather than 'due to alongshore-variable sandbar depths' I think it would be more accurate to say something like 'alongshore variability in depth between the sandbars and intervening channels'

* Line 52: Change 'increased understanding in rip current dynamics' to 'increased understanding of rip current dynamics'

* Line 69: Change 'quantitative estimate' to 'quantitative estimates'

* Line 81: Change 'rip current are ubiquitous' to 'rip currents are ubiquitous'

* Figure 1: The text within panel (b) doesn't show up well unless you zoom in on it. I would suggest changing the text to another colour. In the caption below the figure the abbreviation 'SMGBL' should be spelt in full on it's first use. This would also make it more consistent with the previous photo credit

* Line 103: Change 'operate surf-zone hazard forecast' to 'operate a surf-zone hazard forecast'

* Lines 103-104: Change 'we used numerical wave hindcast' to 'we used a numerical wave hindcast'

* Line 133: Change 'the deeper channel' to 'the deeper channels'

* Line 148: Change 'consider simple' to consider a simple'

* Lines 155-156: U, V, and w should all be defined here (currently only V is defined). Also, please specify where h should be defined as you have a depth over the bar and a depth in the channel. Which is this h supposed to represent?

*Reply : U is not used anymore, and hc and hb are now defined in both the text and Figure 3*

* Line 159: change 'proceeds as follow :' to 'proceeds as follows:'

* Line 160: you should define h_b and h_c here

* Figures 3 and 4: the x and y axes of (d) are not clearly defined. Also, gamma appears here as Y which on first reading seems like a new parameter.

> *Reply : Thank for for this comment, which is in line with a comment of the other reviwer. Figures 3d and 4b have been revised to provide clear insight into the wave height decay model, which arte now of the form (\gamma police was also changed for clarity) :*

[Figure]

* Line 167: Change 'for break type' to 'for breaker type'

* Line 171 and equation 6: Parameter names change from Essb and Hsb here to Esb and Hssb below. Please check all parameter names are consistent.

* Line 180: it is not clear where this wave height is being defined. I assume this is breaker height at the sandbar? If so, it would be clearer and more consistent to refer to this as Hsb (as per earlier definition for rips)

* Line 181: Change 'model proceeds as follow :' to 'model proceeds as follows:'

* Line 190: Change 'Hsl' to 'SHl'

* Figure 6 caption: Change 'pdeth' to 'depth'

* Line 228: Change 'RHl' to 'SHl'

* Line 264: Change 'perception can influenced' to 'perception can be influenced'

* Figure 11 caption: the terminology of 'shore-break wave intensity I' is inconsistent with the parameter naming used up to this point (Esb)

* Line 305: Change 'are modified based a the quantile-quantile' to 'are modified based on the quantile-quantile'

* Figure 12 caption: Change 'with the blue (dotted) blue lines' to 'with the solid (dashed) blue lines'

* Line 323: Change 'rip current tends' to 'rip currents tend'

* Lines 330-331: Change 'it provides a direct Information on' to 'it provides direct Information on'

* Line 351: 'allow to compute the time evolution' - I think it would be fairer to say ' allow to estimate the time evolution'

---

## Author Comment (AC2)

**Overview:**

* This is a well-written paper that is a significant contribution to forecasting rip-current and shore-break hazards using simple models informed by physics and calibrated with lifeguard observations of hazard levels. The introduction is thorough and logically organized and helpful physical parameter schematics are provided. Visualizations throughout are high quality. Time series of physics-informed parameterizations show remarkable agreement with lifeguard assessments of hazard. I found the idealized analysis showing how the models can be applied to hypothetical conditions to be interesting and informative.

> *Reply : We thank Reviewer #2 for their support for publication, constructive criticism, and insightful comments on the model. In the detailed response below, you will see that all the comments have been carefully considered and the necessary changes have been made. We are grateful to the reviewer for having helped us to strengthen our paper.*

* Prior to publication, I think the paper needs to provide more clear derivations and justification of assumptions leading to the new physics-based models; these could appear concisely in the main text or in a more detailed form in supplementary materials. There may be some errors in the rip-current speed and shore-break energy formulas, but it is difficult to assess without seeing more detail in how the authors reached those results. The rip-current hazard formulations based on rip-current speed have previously been derived and compared with lifeguard observations, and the authors derive their result from momentum balances (though more justification is needed). In contrast, the authors note that no theoretical estimate for shore-break hazard yet exists. The proposed shore-break formulation – the product of the Irribarren number and the wave energy – seems highly valuable, but given that it is somewhat ad hoc, maybe it would be more accurately described as semi-empirical or physics-informed rather than physics-based.

> *Reply : We warmly thank Reviewer #2 once again for their insightful comments on the two odels. In the detailed replies to the comments below, you will see that the models have been modified in line with the reviewer's suggestions. While these changes did not alter the overall results and outcomes, except for adjustments to the threshold values and slight modifications to the correlation and confusion matrices (by less than 5%), the construction is now more robust. Additionally, we now systematically refer to the two models as 'semi-empirical' instead of 'physics-based.' The modifications are outlined in our replies below.*

* The second broader comment I have is that some additional discussion of the limitations of this approach and its applicability to other sites would be helpful. Specifically, this approach seems to apply to sites where channel rips dominate, and the importance of other rip current types should be discussed. In addition, for applicability to other sites, it would be good to discuss how a minimal set of sandbar and beach profile shape parameters could be observed directly or estimated through tuning/calibration with lifeguard data, so that readers can assess feasibility. Line-by-line comments below indicate specific places where I suggest clarification on the physics-based parameterization and limitations/applicability.

> *Reply : Thank you for emphasizing this limitation. This is now addressed in the Discussion sections such as "his study focused on channel rip currents, the most common rip type on intermediate beaches, although other types of rip currents exist (see Dalrymple et al., 2011; Castelle et al., 2016b; Houser et al., 2020). With the notable exception of Casper et al. (2024), who explored the potential for forecasting flash rip hazards at a Californian beach, hazard forecasting for other rip current types has never been tested. Our model is therefore mostly adapted for intermediate, high-energy, sandy beaches" Or, when dealing with the replicability of the approach and calibration : "However, while parameters such as bar crest*

*depth and channel depth are relatively simple, obtaining them remains challenging due to the difficulty of surveying the surf zone, which is not routinely monitored at most locations. This raises important considerations for the large-scale transferability of the models. Future applications will need to determine how these parameters can be feasibly obtained, whether through direct surveying, remote sensing, or empirical estimations based on regional morphology. Additionally, while the calibrated values used in this study may serve as a reference, their applicability to other sites remains uncertain, and further research is needed to assess whether re-calibration against lifeguard observations or other validation datasets is necessary at each new location." See also some specific replies to the comments below.*

**Line-by-line comments:**

• L36-38: "The most common rip type" - Clarify, this may be true on some beaches but not others

> *Reply : We agree, we now specify that it is true on intermediate beaches "The most common rip type on intermediate beaches (Wright and Short, 1984; Castelle and Masselink, 2023) flows through channels carved into nearshore sandbars (e.g. Houser et al., 2013)."*

• L73: "The proposed framework offers new opportunities for forecasting rip-current and shore-break wave hazards at surf beaches with available wave predictions" - Morphology information also is needed, and ideally lifeguard observations for calibration. Consider adding these factors to the sentence.

> *Reply : We agree, based on your comment and that of Reviewer #1 this now reads "These simple semi-empirical models providing quantitative estimates of rip-flow speed and shore-break wave energy, and an associated 5-level scale hazard rating, only require a limited number of time-invariant free parameters related to beach morphology and wave breaking onset. These parameters can either be given based to some knowledge of the beach morphology, or through calibration using lifeguard-perceived hazard data. The proposed framework, here applied to a single beach in southwest France, offers new opportunities for forecasting rip-current and shore-break wave hazards at surf beaches with available wave and tide predictions."*

• L131: "Rip current hazard can be estimated through the rip flow speed." Discussion section should cover how flow patterns and other factors may also affect hazard.

> *Reply: Thank you for raising this important point. This is now addressed in the Discussion section "It must also be acknowledged that the rip current hazard in this study was estimated based solely on rip-flow speed. However, other flow characteristics can also influence the physical hazard, such as the rip current circulation regime, which plays an important role in determining the optimal rip-current escape strategy (McCarroll et al., 2014a). Surf-zone rip currents have long been perceived as narrow flows extending well beyond the breakers, rapidly flushing water out of the surf zone in what is known as the 'exit flow' circulation regime. However, studies using Lagrangian drifter measurements to compute surf-zone exit rates (e.g. MacMahan et al., 2010; McCarroll et al., 2014b) have shown that rip-flow patterns can also form quasi-steady, semi-enclosed vortices that retain most floating material within the surf zone, referred to as the 'circulatory flow' circulation regime. Unlike the exit-flow regime, the circulatory regime increases the likelihood that a swimmer caught in a rip current will be carried back to shallower, safer waters within a few minutes (McCarroll et al., 2015; Castelle et al., 2016a). Although observed and modelled exit rates in channel rips show*

*considerable natural variability, the highest exit rates are generally associated with the lowest incident wave energy, and consequently, the lowest rip-flow speeds (see review in Castelle et al., 2016b). "*

• L141, L146: "S=0.16*Hs", "Sb=0.16*Delta-Hsb", "Sc=0.16*Delta-Hsc" Please clarify under what assumptions these approximations are reasonable to use, and what assumptions are involved to modify the approximation for shoreline setup (as a function of wave height) to estimate setup immediately onshore of the bar and channel (cross-shore change in waveheight)? Does this assume breaking in the channel as well as on the bar? My intuition would say that Sb-Sc would then be independent of the offshore wave height, but the squared wave heigh decay equation suggests otherwise (see next comment). How does this more simplified approximation compare with other formulations that include more parameters, e.g., Moulton et al. 2017 / Casper et al. 2024? A simpler formulation with fewer parameters is ideal for hazard prediction if it is clarified under what conditions it is a reasonable approximation. It seems like this formulation could be roughly a factor of 4 larger than Moulton/Casper, but I'm not completely sure, especially given the complexity of the quadratic delta-H formula.

> *Reply : Thank you for pointing this. The assumption for obtaining Sb and Sc (note that variable names were changed according to a later comment) are now clarified : "Considering Equation (1), but looking immediately onshore of the bar/rip system instead of the waterline, where the entire incident wave energy has been dissipated, and by further ignoring set-down, wave refraction, wave-current interaction, we can make the first-pass assumption that wave-set up immediately onshore of the bar/rip system is controlled by the change in wave height due to depth-induced breaking across the bar and/or the channel. We can therefore assume ηb = 0.16ΔHsb and ηc = 0.16ΔHsc, where ΔHsb and ΔHsc are the decrease in wave height due to depth-induced breaking across the bar and the channel, respectively (Figure 3b)". Sb-Sc would be independant on wave height if (1) breaking occurs in across both the channel and the bar and (2) if only regular waves are considerd (see response to nexty comment), which is a major difference with Moulton/Casper approach. The advantage here is, as Reviewer #2 indicates, that our formulation is simpler, but the drawback is that physical fundations of the approach is less robust because we do not consider regular waves (see also next comment). We think that the two approaches are complementary, and in the end provide similar results. Future work could incolve coomparing the two approaches.*

• L148-151: "Here we consider simple first-pass estimation of the significant wave height decay for irregular waves." – Is there a reference for this? Or provide a derivation or more explanation. Assuming a wave breaking gamma and single wave height, I would expect Delta-Hs to be simply Hs-H, where H=gamma*h for broken waves. Does Equation (2) differ from this due to considering an irregular wavefield, e.g., Rayleigh distributed wave heights? & L150: It could be worth spelling out the two equations for Hsb and Hsc, so that the dependence of the speed on the bar-channel geometry is clearer

> *Reply : Thank you again for pointing this out, and we apologize for not providing enough information about the underlying assumptions. Equation (2) indeed differs from the simple formulation Hs=γh because we are considering random waves. Unfortunately, unlike regular waves, there is no straightforward solution for estimating irregular wave heights in the surf zone, even for planar beaches. Therefore, this simplified approach (Equation (2) and Fig. 3d) was inspired by the typical cross-shore distribution of Hs or Hrms observed in the surf zone for irregular waves and planar beaches. In reality, this distribution depends on many other factors, but it serves as a first-order, simplified approximation to the known curves (see Dally, 1990, Coastal Engineering). This distribution also explains why Sb−Sc depends on offshore wave height. We have now clarified this in the model description section, explicitly stating the*

*two equations for Hsb and Hsc. "Critical to both ΔHsb and ΔHsc is the depth-induced breaking wave height decay law. Unlike regular waves, there is no simple method to estimate irregular wave heights in the surf zone, even on planar beaches. Previous studies (Dally, 1990) have shown that the root mean square wave height distribution in the surf zone on planar beaches depends on various factors, including beach slope and wave steepness. However, by neglecting wave shoaling effects and for the sake of simplicity, a physics-informed (Dally, 1990) estimation of the depth-induced breaking significant wave height decay, ΔHs, for irregular waves (Figure 3d), can be expressed as:*
*Eq(2)*
*for hi > 0 and Hs > γhi (broken waves), where hi is the local water depth with subscript i referring to the bar (i = b) or the channel (i = c), γ is the breaker parameter for random waves, and Hs is the significant wave height at breaking (after transformation through Larson et al., 2010). The depth-induced breaking significant wave height decay over the sandbar ΔHsb (the channel ΔHsc) are given by:*
*Eq (3)*
*Eq (4)*
*with ζ the tide elevation, zbar the elevation of the sandbar and d the channel depth (Figure 3b)."*

• L152: Please provide references and/or justification for the simplified momentum balance & L155: I think more justification is needed for these approximations. Is it known that the setup varies over a lengthscale of the width of the channel? Why not a half-width, or a multiple of the width, or something else like the spacing between channels, or a frictional lengthscale? I don't think this is actually known. Similarly, for the advective term, given the argument is that this is a physics-based parameterization, a derivation should be provided. Using the continuity equation with the left-hand side of Equation 3, it is not clear how the 2*V^2*h/w approximation is reached. Are assumptions made about U=V or U=1/2*V or U=2*V? Is the alongshore lengthscale w or ½*w or 2*w? Is it assumed that alongshore depth variations are small (dh/dx * 1/h is small)?

*Reply : We thank Reviewer #2 for raising this point. For consistency we now use the same approach to link rip flow velocity and gradients in wave set-up. This reads "Following Moulton et al. (2017b), we assume that the ratio of bottom stress to the advection term is small, and that the balance of pressure gradients and advection along a streamline can be approximated using the Bernoulli equation. By further neglecting the effects of inertia in a longshore current driven by obliquely incident breaking waves, the rip flow velocity V can be approximated as:*
*Eq (5)*
*where $\eta_b = 0.16\Delta H_{sb}$ and $\eta_c = 0.16\Delta H_{sc}$ the wave set-up onshore of the bar and of the channel, respectively. Note that, because of the irregular wave height decay law (Equation (2)), the alongshore gradient in wave set-up, and thus rip-flow speed V , depend on d, zbar and Hs, whereas assuming regular waves, it would be independent of Hs when depth-induced breaking occurs both over the channel and the sandbar."*

• L157: (Equation 4) I'm not convinced this formula is correct. The Moulton 2017 / Casper 2024 formula would be sqrt(2*g*(Sb-Sc)), which is different from this by a factor of 2. The Sb-Sc formula may have an extra factor of 4 relative to the Moulton 2017 setup difference estimate. Interestingly, these differences would compensate each other. I would have most confidence in a formulation that is consistent with past work that has been compared with field observations of speeds.

*Reply : This was corrected (see previous comment). Please note that we recomputed all the rip current outputs and modified the figures and table accordingly. Given the proprotionnality between former and revised rip flow speed velocities, results did not change, only the thresholds did.*

• Figure 3: The way S(x) is drawn as a square wave, dS/dx is not differentiable… would it make sense to show linear variations in S from the bar to the channel center instead?

*Reply : Thank you for this comment we now shown a linear alongshore variation of wave set-up*

• Figure 3d, 4b: I am confused by the diagrams in Figure 3d and 4b. What are the x and y axes?

*Reply : Thank for for this comment, which is in line with a comment of the other reviwer. Figures 3d and 4b have been revised to provide clear insight into the wave height decay model, which arte now of the form :*

[Figure]

• L165: "no theoretical framework to estimate a measure of the shore-break wave energy" – If this is the case, I might describe the following formulations as physics-informed rather than physics-based, but this is a wording nuance

*Reply : We fully agree with this comment. Because depth-induced wave breaking dissipation model used for both the shore-break and rip current models is physics-informed, we now use "semi-empirical" instead of physics-based throught the manuscript. Please note that the paper title was slightly reworded accordingly accordingly.*

• L186: Does the squared quantity come from the same "decay law" used in the rip-current formulation? Could write this as a 3-part equation for wave-breaking types (subaerial bar, bar-breaking, and shoreline-breaking)?

*Reply : The Reviewer is right, this is now clarified "..., following the same depth-induced breaking irregular wave height decay law as for the rip current model, ..."*

• L188: Could Z_l be written as z_bar, for consistency with the rip-current formula?

*Reply : We agree, it was replaced by in the text and figures*

• L169: "deep water wavelength" - Is it possible that the wave condition upon shore-breaking deviates from the deep-water wavelength, since breaking on the bar could filter out some frequencies given differences in steepening and breaking? Particularly for wavefields with broad or multi-peaked frequency spectra. Could you comment on when using offshore wavelength is relevant?

> *Reply : this is now specified "herefore, we introduce a shore-break wave energy parameter $E_{sb} = I_r H^2_{sb}$, assuming no change in peak wave period as the waves pass over the sandbar(s) before reaching the shore, which therefore reads : ..."*

L169: Should this be $T_p$ squared?  L173: Is a factor of sqrt(2*pi) missing in the equation?

> *Reply : The Reviewer is right. While it was a typo for $T_p$ squared, it was an omission for $2*\pi$. Although this does not changes the message of the paper and overall results, we recomputed the time series and changed the figures accordingly. Model skill only marginally changed.*

• L195: "thresholds were computed in order to obtain the same number of modelled hazard levels" – does it need to be exactly the same number? You could allow some uncertainty to avoid overfitting / specify confidence intervals on this choice of ranges. I doubt the confidence is reflected in the significant digits shown, with 1 cm/s and 0.01 m^2 resolution.

> *Reply : This was to our knowledge the only approch possible to use without using some a priori assumption. We understand that some confidence band could be provided, but the limited number of data available, especially in the high hazard range, does allow us to do so.*

• L200: "daily-mean" – Is the mean, max, or median most relevant for hazard? I would think maximum may be most relevant. Daily is somewhat coarse. I wonder about having at least having morning and afternoon to capture some of the tidal variability, and could be relevant for shift staffing by lifeguards.

> *Reply : This is a relevant comment, and it is also why we believe that the hourly and coarser daily mean approaches are complementary. Lifeguards in France are deployed on a daily basis, which makes the daily approach particularly relevant. We have now clarified why and how these two approaches complement each other, and we explain how the daily approach can also be useful in this context : "The daily-mean rip-current hazard forecast is important for providing a straightforward message to the general public, and can also assist lifeguard managers in scheduling lifeguards in advance, ensuring they are deployed to the beaches where they will be most needed. In this context, the daily-mean wave factor ($W_f$) appears to be a simple yet powerful tool for predicting and communicating high rip-current hazard days. It is also important to note that the correlation between the hourly lifeguard-perceived rip current hazard ($RH_l$) and the hourly wave factor ($W_f$) remains relatively high ($R = 0.65$). This indicates that, although $W_f$ alone does not account for tidal modulation, it still explains more than 40% of the observed variability in lifeguard-perceived rip current hazard. Overall, predicting daily-mean $W_f$ is complementary to the higher-frequency rip-current hazard hourly prediction throughout the day with our semi-empirical model, and to the shore-break hazard model which can be used for both daily-mean and hourly predictions."*

• L208-211: "by merging […] into low-hazard […and…] moderate- to high-hazard hours […], the accuracy increases" – It would be worth discussing here or in the Discussion why the 5-level scale did not perform well. Was it because there wasn't enough data or that the

parameterizations capture a clear enough relationship between inputs and outputs to predict hazard on a finer scale?

*Reply : Mechanically, the accuracy in the confusion matrix increases with the number of levels in the hazard scale. We already discussed why excellent metrics cannot be reached here, notably because here we are using lifeguard-perceived data, we now expand on this in the revised manuscript, including using other sources of data : "Indeed, as beach safety professionals, lifeguards are supposed to develop a more robust hazard perception than laypersons (Sandman et al., 1987; Slovic, 1999). However, according to Rowe and Wright (2001), it can also be argued that lifeguards remain human beings whose hazard perception can be influenced by personal factors (experience, gender, etc.). Using average lifeguard-perceived hazard data from all the lifeguards on duty, instead of the chief lifeguard only, could provide a more robust data to calibrate the model. The validation approach proposed here can be applied anywhere pending lifeguard hazard assessment can be performed. If such lifeguard data cannot be collected, a first-pass approach is to base the hazard level scales on the threshold values computed in southwest France (Table 1). Once again, such model application together with lifeguard-perceived hazard should be tested elsewhere to address the influence of beach state, modal wave climate and lifeguard perception on these threshold values. Since collecting consistent hourly lifeguard-perceived hazard data over a few weeks and under varying tide and wave conditions may not be feasible at many locations, an alternative approach is to use lifeguard-reported incidents (see, for instance, Scott et al., 2014). While such data also incorporate the exposure component of risk (Stokes et al., 2017), they are more widely available and can be highly valuable, particularly for assessing whether the model can identify mass-rescue days."*

• L230: "outliers" – Might these be worth discussing further since hazardous events that are "outliers" and not well forecast could be dangerous.

*Reply : we think that outliers may also be because the training dataset is not perfect, see reply to previous comment*

• Figure 6,9: Since panels a and b are duplicated in these two figures, consider merging these in one figure with both of the full the rip current and shore-break time series, which may be interesting to show how they vary differently with conditions (similar to Figure 11). The example shorter time window in panels d-i could be two separate figures for rip currents and shore-break. Just a suggestion.

*Reply: Thank you for this suggestion. We conducted some tests, but merging the two figures did not allow for providing zoomed-in plots, which we believe are useful.*

• Figure 7,10: Would a bin average help to show if the model tends to be over- or under-forecasting at different hazard levels?

*Reply : We believe this is already addressed in the right-hand panels and in the hourly confusion matrices (Figures 5 and 8).*

• L269: "should be tested elsewhere" – Here or in the Discussion (could go with paragraph beginning on line 295 in the Discussion), it would be good to discuss how the sandbar elevation and beach profile shape parameters can be inferred, and/or the need to get these morphology parameters through tuning/calibration with lifeguard data, which is also hard to get. In addition, note that this approach assumes that the beach is always channeled, and that channel rips are the strongest rips, as opposed to transient rip currents, structural rips, etc.

*Reply : Thank you for this comment, this is now discussed : "However, while parameters such as bar crest depth and channel depth are relatively simple, obtaining them remains challenging due to the difficulty of surveying the surf zone, which is not routinely monitored at most locations. This raises important considerations for the large-scale transferability of the models. Future applications will need to determine how these parameters can be feasibly obtained, whether through direct surveying, remote sensing, or empirical estimations based on regional morphology. Additionally, while the calibrated values used in this study may serve as a reference, their applicability to other sites remains uncertain, and further research is needed to assess whether re-calibration against lifeguard observations or other validation datasets is necessary at each new location." We also now precise that we only look at channel rips : "his study focused on channel rip currents, the most common rip type on intermediate beaches, although other types of rip currents exist (see Dalrymple et al., 2011; Castelle et al., 2016b; Houser et al., 2020). With the notable exception of Casper et al. (2024), who explored the potential for forecasting flash rip hazards at a Californian beach, hazard forecasting for other rip current types has never been tested. Our model is therefore mostly adapted for intermediate, high-energy, sandy beaches"*

• L283: "daily-mean lifeguard perceived hazards" - Would daily max be better for hazard preparation, given that the mean could obscure a brief but high-risk time period? Or split into morning vs afternoon max or mean?

*Reply : See reply to a previous comment on the complementarity of daily-mean and hourly forecasts.*

• Figure 12: Why is the Dean profile so different from the measured profile?

*Reply : As discussed, using a more realistic Dean profile only slightly decrease model skill (note that with the sloghtly modifed shore-break wave model the decrease is even smaller) : "he Dean profile (solid blued line in Figure 11b) is much steeper than the alongshore-averaged profile. However, by changing a=-2.75 into a=-1.75, which is in much better agreement with the measured profile (dotted blue line in Figure 11b), the correlation between Esb and SHl is approximately the same (R=0.70, with a marginal decrease by ≈ −0.5% using the dotted blue line profile in Figure 11b). This once again shows that beach surveys can be used instead of a Dean profile calibrated with lifeguard-perceived hazards.*

• L322: "weak but significant" - Is this statistically significant?

*Reply : not really, "significant" has been removed*

• L366: "only a few basic beach morphology metrics" - This may be a little vague and subjective use of "basic," clarify.

*Reply : We agree that "basic" is not the appropriate term here, it has been removed.*

• L299-L305: d=6.5 m seems like an unrealistically deep channel. Did you consider constraining the parameter range in the fit to physically realistic values? The skill was similar for more realistic values so this would not change the results much but could provide more realistic predictions for future conditions.

*Reply : We allowed the model to consider a wide range of morphology metrics. However, constraining the model to more realistic values results in only a slight decrease in model skill,*

*as discussed : "For instance, the correlation between V and RHl decreased slightly from 0.77 to 0.75 (≈ −3%) when assuming a higher bar crest (zbar = −2 m instead of -3 m) or a much shallower channel (d = 2 m instead of 6.5 m), which are closer to average values in southwest France. This suggests that a decent model skill can be achieved with a rough estimate of the bar/rip morphology, further implying that temporal variability in beach morphology can be neglected in the model."*

**Minor typographic suggestions:**

*Reply : all the minor suggestions liste below were take into account, unless specified below. Please note that these comments are mostly related to colors. We thank Reviewer #2 for their interest for making scientific results accessible to everyone, including color blind people. Interestingly enough, the 1st author of this paper is colour blind, at a severe stage which, I can testify, is a real pain when dealing with plots from students, papers, talks in conference, etc. Surprisingly, I am severely colour blind, and jet is still kind of my favourite as it goes through the 3 primary colors, which is great compared to some proposed colorbars which are terrible (at least to me). In fact, Jet colorbar is only terrible for achromatopsia, which is the most severe and rarest colour blindness, and not very good for the blue-yellow vision deficiency, which is pretty rare too. But it is still pretty good, and to me one of the best, for the red-green deficiency of which I have more or less the 4 sub-types of deficiency. I have tested many "color-blinf friendly" colormaps but they are much worse than Jet for me, I actually see not much contrast. Instead jet colormap or any plot where primary colors are used are the only way for me to see contrast. Given that I want to use these figure for presentation and be able to easily see the contrast I therefore decided to stick to jet colorbar and primary colors in the revised manuscript. I however want to thank again Reviewer #2 for their interest of data visualisation for color blind people.*

§ L2: change "can expose to" to "can be exposed"

§ L5: change "allow to compute" to "can be used to compute the time"

§ L8: August date missing, August 31?

§ L12: change "where wave forecast is available" to "where wave forecasts are available"

§ L14: remove "e.g."

§ L17: remove "been" in "have been greatly increased"

§ L18: remove "e.g."

§ L43: lengthy paragraph- could start a new paragraph at "Shore-break"

§ L69: change "estimate" to "estimates"

§ L70: add "an" before "associated 5-level scale"

§ L71: change "can be either given thanks to" to "either be given based on"

§ L71: remove "e.g."

§ L103: change "surf-zone hazard forecast" to "surf-zone hazard forecasts"

§ L104: change "numerical wave hindcast instead" to "numerical wave hindcast data instead"

§ L104: change "consisted in an analysis" or "consisted of an analysis"
§ L108: change "as unified" to "including unified"

§ L111: change "was assumed representative" to "was assumed to be representative"

§ L126: here "RH_l" and "SH_l" have a "l" subscript- correct typos throughout the manuscript where "l" is not subscripted- same comment for "m" subscript

§ L135: notation- consider using x as cross-shore coordinate and y as alongshore coordinate for consistency with most of the surfzone literature, also consider eta-bar for setup instead of S, which is typically used for radiation stress

> *Reply : We kept the x/y coordinates system by replaced S by eta-bar, and the tide elavation was changed from eta to zeta. Modifications have been made in both the text and figures*

§ Figure 3: Hard to see the red text on the blue background in panel a.

> *Reply : See colorblind comment*

§ Figure 4: Consider changing the notation for the terrace elevation, Z_l, elsewhere "l" is used for lifeguard.

§ L171: Should the subscript be ssb for H?

§ Figure 5,8: is it typical for the y axis to be flipped like this in the confusion matrix?

> *Reply : Yes it is*

§ Figure 11: hard to see the difference between the dark pink and red, could switch to red-blue colors in panel d or other colorblind friendly palette

> *Reply : See colorblinf comment*

§ L285: add "indicating" after "notation"

§ Figure 12: consider switching to a colorblind friendly colorbar, caption unclear- suggest rephrasing to "with the blue solid (dotted) lines depicting…"

> *Reply : see colorblind comment*

§ L332-L334: remove unnecessary use of "e.g."

§ L336: change "model" to "models"